# On the Fragility of Data Attribution When Learning Is Distributed

Xian Gao [1]   Bo Hui [2]   Min-Te Sun [3]   Wei-Shinn Ku [1]

## Abstract

Data attribution has become an important component of pricing, auditing, and governance in machine learning pipelines, yet most attribution methods implicitly assume that attribution values faithfully reflect participants' contributions. We show that this assumption can fail: a single participant in a standard distributed training workflow can substantially inflate its measured attribution value while preserving global utility. Our attribution-first attack uses latent optimization to inject small synthetic batches that preserve utility while exploiting non-IID label coverage and evaluator sensitivities. Across datasets, models, and multiple marginal-utility evaluators, the attack consistently increases the adversary's attribution value and reshapes the relative attribution structure among benign clients without degrading accuracy or triggering geometry-based defenses. These results show that attribution itself forms a new attack surface and motivate the development of attribution-robust and incentive-compatible scoring mechanisms.

## 1. Introduction

Modern machine learning increasingly depends on data attribution, which aims to quantify how individual samples or data owners influence model behavior and utility. Recent work has advanced data valuation from multiple perspectives, including Shapley-style marginal-utility estimation (Chen et al., 2023; Jiang et al., 2023; Liu et al., 2023; Li & Yu, 2023) and influence-function or optimization-based approximations (Nguyen et al., 2023; Bae et al., 2024; Covert et al., 2024a). These methods support emerging applications such as data pricing (Zhang et al., 2025a), governance of collaborative training systems (Murhekar et al., 2024), and auditing of modern vision and language models (Wang et al., 2024a). In practice, a hospital consortium may allocate reimbursement based on each site's contribution to rare-condition sensitivity; a platform may reward data cooperatives whose logs improve long-tail categories; and a model marketplace may admit or remove suppliers using attribution thresholds (Kandpal & Raffel, 2025; Murhekar et al., 2023; Hesse et al., 2024). As attribution becomes more tightly integrated into trustworthy AI pipelines (Deng et al., 2024; Covert et al., 2024b), maintaining the integrity of attribution scores becomes increasingly important for system reliability.

Collaborative training frameworks such as federated learning (FL) raise important challenges for data attribution. While raw data remains local to each client, attribution metrics are increasingly used to assign credit, compensation, and accountability across participating institutions (Song et al., 2019; Fan et al., 2024). This intersection between attribution and FL is particularly relevant in cross-silo settings, such as multi-institution healthcare collaborations, where data sharing is restricted but joint training is necessary. Recent federated systems and incentive frameworks have therefore explored contribution-aware mechanisms for participant valuation, fairness, and reward allocation (Murhekar et al., 2023; Chen et al., 2023; Tastan et al., 2024). At the same time, FL deployments often involve heterogeneous, naturally fragmented, and highly non-IID data (Yang et al., 2023; Wang et al., 2024c), creating structural imbalances that attribution rules attempt to correct. Recent work on data valuation shows that even small changes in local coverage can disproportionately affect marginal-utility estimates (Xu et al., 2024; Covert et al., 2024a; Sun et al., 2024b). These observations raise a natural but *underexplored* question: can a participant strategically shape its updates to appear unusually beneficial without actually harming global utility?

Existing threat models and defenses in FL are largely model-centric, focusing on poisoning attacks (Bal et al., 2025; Pawelczyk et al., 2024) or anomalous updates (Jia et al., 2024), rather than attribution manipulation. Recent adversarial work on data attribution (Wang et al., 2024e) studies inflation of per-sample data value in centralized settings, but does not address client-level contribution manipulation

[1]Department of Computer Science and Software Engineering, Auburn University, Auburn, Alabama, USA [2]Department of Computer Science, University of Tulsa, Tulsa, Oklahoma, USA [3]Department of Computer Science and Information Engineering, National Central University, Taoyuan, Taiwan. Correspondence to: Bo Hui <bo-hui@utulsa.edu>.

*Proceedings of the 43rd International Conference on Machine Learning*, Seoul, South Korea. PMLR 306, 2026. Copyright 2026 by the author(s).

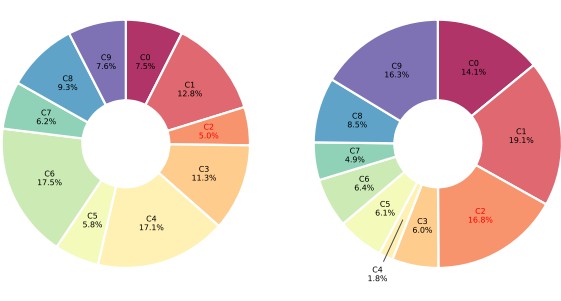

*(a)* Attack-Free    *(b)* Latent Optimization Attack

*Figure 1.* Client-level attribution can shift under utility-preserving local changes. Attribution shares for an attack-free (left) versus a latent optimization attack (right).

in FL. At the same time, recent studies on data valuation have documented instability, hyperparameter sensitivity, and evaluator variance (Wang et al., 2025c; Wei et al., 2024; Wang et al., 2024b; Rubinstein & Hopkins, 2025), suggesting that attribution signals can be surprisingly malleable in practice (Wang et al., 2025a). Together, these observations point to a critical gap: although FL increasingly depends on attribution for incentives and governance, the attribution pipeline itself remains largely unprotected.

In this paper, we take an attribution-first security perspective and investigate whether a single participant in a standard multi-party training workflow can inflate its measured contribution by exploiting non-IID label distributions, evaluator sensitivities, and benign-looking local updates without affecting model utility. Our perspective is motivated by recent work showing that marginal-utility estimators depend strongly on coverage, curvature, and model dynamics (Wang et al., 2023; Lin et al., 2024c; Sun et al., 2024a; Mlodozeniec et al., 2026). Our preliminary study shows that mild and plausibly benign adjustments to a client's synthetic training batch, optimized only through the broadcast global model, can produce persistent and disproportionately large attribution gains. For example, Figure 1 shows that a single client can induce large attribution shifts through mild local modifications while keeping test accuracy within a small tolerance. In particular, when Client 2 (C2) behaves adversarially, its attribution value increases substantially, while the attribution shares of most benign clients decrease. At the same time, a small subset of benign clients shows unchanged or slightly increased attribution, indicating that the attack perturbs the relative attribution structure rather than uniformly shifting all attribution values. Crucially, these attribution distortions arise without harming model utility, as standard performance metrics such as test accuracy remain stable within normal variance.

The implications extend beyond the specific attack studied here. As large-scale collaborative training expands across modalities and architectures (Kwon et al., 2023; Zheng

et al., 2023; Wang et al., 2024d; 2025b), attribution metrics increasingly shape economic incentives, participation rights, and provenance guarantees. If attribution can be manipulated without harming global performance, systems may reward the wrong contributors, misidentify harmful actors, and draw misleading conclusions about data quality (Murhekar et al., 2023; Jiang et al., 2023; Li & Yu, 2023). Maintaining model accuracy alone is therefore insufficient; attribution integrity is also necessary for trustworthy collaborative learning. To study this issue, we design an attribution-aware attack that operates entirely within the standard FL workflow. Rather than degrading model accuracy or injecting anomalous updates, the attacker introduces only mild, utility-preserving modifications to its local training process. By exploiting non-IID label coverage and the sensitivity of attribution evaluators to marginal utility, a single client can systematically amplify its measured contribution while remaining statistically plausible under common sanity checks. This attack exposes a fundamental vulnerability in attribution pipelines: even when global performance remains stable, attribution signals can still be strategically manipulated.

We summarize the contributions as follows:

- **Attribution-first threat model.** We formalize a new attack objective in FL that inflates a client's data-attribution score under explicit utility and plausibility constraints.

- **Practical latent-optimization attack.** We propose a practical attack that injects a small amount of decoder-generated synthetic data to construct attribution-aware local updates that remain utility-preserving and statistically plausible.

- **Theory and empirical validation.** We provide theoretical insight into why such updates increase marginal attribution without degrading accuracy, and empirically validate the attack across datasets, models, and attribution evaluators.

## 2. Problem Formulation

### 2.1. Threat Model

**Attacker's Capabilities.** Beyond standard FL assumptions, we consider a single malicious client that can (i) construct and inject a small amount of on-device synthetic data into local training to shape the reported update, and (ii) infer the attribution protocol class used by the server, since such information is often publicly documented or can be inferred from repeated observations. The attacker observes the broadcast global model $w_t$ at each round and may cache limited history for stability. In particular, the attacker constructs a benign reference direction from observable global model differences $g_t^\dagger \approx w_t - w_{t-1}$ without access to other clients'

updates. The attacker only uses its own local dataset $\mathcal{D}_r^{(i^\star)}$ and does not access other clients' private data, consistent with the standard cross-silo FL setting. We assume the attacker has access to a fixed pre-trained decoder $\text{Dec}(\cdot)$. Such decoders can be obtained from public models or lightweight on-device distillation and are used only for inference, introducing negligible overhead compared with standard local training.

**Attacker's Limitations.** The attacker cannot modify server-side aggregation or evaluation, nor interfere with other clients. Crucially, the attack must preserve global utility $U(\cdot)$ (e.g., test accuracy). Let $U^{(0)}$ and $U^{(1)}$ denote the final utilities of the clean (no-attack) run and the attacked run, respectively. Then

$$\left| U^{(1)} - U^{(0)} \right| \leq \delta, \tag{1}$$

for a small tolerance $\delta$, and all synthetic samples must lie within the valid task domain $\mathcal{X} \times \mathcal{Y}$ (where X represents inputs and Y represents labels).

### 2.2. Formal Attack Objective

Let $w_t$ be the global model broadcast at round $t$, and let $E(\cdot)$ be the server-side attribution evaluator. We focus on a single malicious client $i^\star$. Denote by $g_t^{i^\star}$ the benign local update induced by the attacker's own real dataset $\mathcal{D}_r^{(i^\star)}$, and by $\hat{g}_t^{i^\star}$ the reported update under attack. Let $C(\cdot)$ be the communication cost, $\Gamma$ a reference set of benign updates, and $C_{\max}, \epsilon, \kappa > 0$ the system budgets.

The attacker aims to increase its attribution score while preserving global utility. Each round it selects

$$\hat{g}_t^{i^\star} \in \arg\max_g E(g) \quad \text{s.t.} \quad \text{Eq. (1) holds.} \tag{2}$$

Equivalently, the goal is to maximize the marginal gain $E(\hat{g}_t^{i^\star}) - E(g_t^{i^\star})$ while keeping utility within tolerance $\delta$.

To avoid trivial detection, the reported update must satisfy:

$$
\begin{aligned}
&\text{(budget)} &&C(\hat{g}_t^{i^\star}) \leq C_{\max}, \\
&\text{(plausibility)} &&d(\hat{g}_t^{i^\star}, \Gamma) \leq \epsilon, \; \|\hat{g}_t^{i^\star}\| \leq \kappa, \\
&\text{(domain)} &&\mathcal{D}_s \subseteq \mathcal{X} \times \mathcal{Y},
\end{aligned}
\tag{3}
$$

where $\mathcal{D}_s$ is the synthetic dataset used by the attacker, $d(\cdot, \Gamma)$ measures deviation from the benign-update reference set $\Gamma$, and $C_{\max}, \epsilon, \kappa$ bound the communication cost, deviation, and update norm, respectively.

Across $T$ rounds, the attacker maximizes its cumulative attribution gain:

$$\max_{\{\hat{g}_t^{i^\star}\}_{t=1}^T} \sum_{t=1}^T \left( E(\hat{g}_t^{i^\star}) - E(g_t^{i^\star}) \right) \tag{4}$$
$$\text{s.t.} \quad \text{constraints in Eq. (3) and Eq. (1).}$$

This formulation isolates what is optimized—attribution gain under utility and plausibility budgets—from how updates are produced, which is realized through the latent optimization attack in Section 3.2.1. Although the formal objective is expressed as maximizing the server-side evaluator $E(g)$, this quantity is not directly optimizable by a client. Instead, the attacker optimizes a latent surrogate objective that captures the structural preferences of $E(\cdot)$ while preserving global utility. This surrogate is implemented via a joint loss function in the latent space, enabling attribution-aware update construction without direct access to the evaluator.

## 3. Latent Optimization Attack

### 3.1. Overview

We study an attribution-manipulation attack where a single malicious client $i^\star$ increases its attribution value without visibly harming global utility. The key idea is to augment benign local training with a small set of decoder-generated samples whose latent vectors are optimized each round to steer the client's update toward directions favored by the server.

In each round $t$, the server broadcasts $w_t$, and client $i^\star$ trains on its real shard together with a compact synthetic batch decoded from a latent vector $z$. The synthetic batch is lightweight and co-evolves with the global model across rounds. Rather than degrading accuracy, the attacker seeks to appear beneficial: synthetic samples fill label-coverage gaps and steer the update toward directions rewarded by attribution evaluators while remaining consistent with benign update norms and shapes. The latent vector is refined each round so decoded samples generate gradients aligned with global progress and plausible under server-side checks. After refinement, the attacker mixes the synthetic and real data, performs a standard local update, and reports it for aggregation. Notably, the synthetic samples need not be photorealistic or distribution-matching in pixel space; alignment is enforced at the gradient level through the supervised loss, ensuring compatibility with the attribution evaluator without requiring domain-level visual realism.

### 3.2. Detailed Latent Optimization Attack

#### 3.2.1. PIPELINE DETAILS

We describe the end-to-end procedure executed by the malicious client $i^\star$ in each communication round $t$. Let $w_t$ denote the broadcast global model and $z$ a latent vector decoded by a fixed generator $\text{Dec}(\cdot)$ into synthetic samples $(\tilde{\boldsymbol{x}}, \tilde{y}) \in \mathcal{X} \times \mathcal{Y}$. The attack consists of five stages.

(1) Client $i^\star$ receives a standard shard of real data $\mathcal{D}_r^{(i^\star)}$, identical to benign clients. To bootstrap synthetic generation, the client samples an initial latent vector $z \sim \mathcal{N}(0, I_d)$,

the $d$-dimensional isotropic Gaussian prior used by the decoder (or warm-starts from round $t-1$).

(2) At the beginning of round $t$, the client refines $z$:

- Generate a mini-batch $(\tilde{x}, \tilde{y}) = \text{Dec}(z)$.
- Choose $\tilde{y}$ from classes underrepresented or absent in $\mathcal{D}_r^{(i^\star)}$ to produce "exclusive" coverage.
- Update $z$ using $\nabla_z \mathcal{L}(z)$, where $\mathcal{L}$ is the joint loss (Sec. 3.2.2). A few gradient descent steps suffice to steer $z$ toward gradients that appear valuable and benign.

(3) After refinement, the client decodes a new synthetic batch and performs local training using both its real samples and the generated synthetic ones. The real portion keeps the update looking benign, whereas the synthetic portion fills in missing labels and steers the update toward directions that yield higher attribution scores.

(4) The client forms the post-training update $\hat{g}_t^{i^\star}$ and verifies that its direction and norm satisfy protocol constraints. Once validated, the update is uploaded to the server for aggregation.

(5) The attacker caches $(z, w_t)$ and uses the refined $z$ as the warm start for round $t+1$. This enables the synthetic data to co-evolve with the global model, allowing attribution gains to accumulate across rounds.

### 3.2.2. JOINT LOSS FUNCTION DETAILS

We optimize a single latent vector $z$ that parameterizes the synthetic batch via the fixed decoder $\text{Dec}(\cdot)$. The objective combines three terms without explicit coefficients:

$$\mathcal{L}(z) = \mathcal{L}_1(z) + \mathcal{L}_2(z) + \mathcal{L}_3(z).$$

The joint loss contains three parts; $z$ is the variable to be optimized.

- $z$: latent vector representing the synthetic dataset $\Rightarrow$ optimization variable.
- $\mathcal{L}_1(z)$: gradient direction alignment $\Rightarrow$ ensures synthetic gradients align with a benign reference direction.
- $\mathcal{L}_2(z)$: gradient magnitude alignment $\Rightarrow$ matches gradient norms to benign clients for stealth.
- $\mathcal{L}_3(z)$: standard task loss $\Rightarrow$ keeps generated data effective for training.

**Direction alignment.** Let $w_t$ be the broadcast global weights at round $t$, $f_{w_t}$ the corresponding model, and let $g_t^\dagger$ be a reference benign gradient in round $t$, e.g., a global descent direction such as the difference between two consecutive global models $w_t - w_{t-1}$. Let $\nabla_w \ell(f_{w_t}, \text{Dec}(z))$ denote the gradient of the task loss on the decoded batch. We align the synthetic-gradient direction with this reference

direction via cosine similarity:

$$\begin{aligned}
\mathcal{L}_1(z) &= 1 - \cos\theta(z) \\
&= 1 - \frac{\langle \nabla_w \ell(f_{w_t}, \text{Dec}(z)), g_t^\dagger \rangle}{\|\nabla_w \ell(f_{w_t}, \text{Dec}(z))\| \, \|g_t^\dagger\|}.
\end{aligned} \quad (5)$$

Symbols used for this term:

- $\text{Dec}(z)$: decoder that generates a synthetic training batch from latent vector $z$.
- $f_{w_t}$: global model at FL round $t$.
- $\nabla_w \ell$: gradient of the supervised task loss with respect to the model weights.
- $g_t^\dagger$: reference benign gradient (e.g., approximated via $w_t - w_{t-1}$).
- $\langle \cdot, \cdot \rangle$: inner product; $\|\cdot\|$: $\ell_2$ norm.

**Norm consistency.** We minimize the gap in gradient norm between the synthetic batch and a benign reference $g_t^\dagger$ (e.g., instantiated via the observable global difference $w_t - w_{t-1}$), which reduces detectability by norm-based defenses:

$$\mathcal{L}_2(z) = \Big| \|\nabla_w \ell(f_{w_t}, \text{Dec}(z))\| - \|g_t^\dagger\| \Big|. \quad (6)$$

**Task fidelity.** To keep decoded samples class-discriminative, we include the standard supervised cross-entropy on the synthetic batch:

$$\mathcal{L}_3(z) = \text{CE}(f_{w_t}(\tilde{x}), \tilde{y}). \quad (7)$$

Here $(\tilde{x}, \tilde{y}) = \text{Dec}(z)$, and labels $\tilde{y}$ are chosen from classes underrepresented or missing in $\mathcal{D}_r^{(i^\star)}$.

Symbols used for this term:

- $\tilde{y}$: ground-truth labels for the synthetic batch.

**Weighting strategy.** All three terms are used with equal weights, reflecting the three structural conditions required to jointly preserve utility while increasing attribution. Using equal weights avoids introducing additional hyperparameters that are not central to the scientific contribution of this work, and we observe stable behavior across datasets, models, and attribution evaluators without tuning. Since our focus is on demonstrating the manipulability of attribution mechanisms rather than optimizing surrogate objectives, a systematic sensitivity study of the weighting scheme is left to future work oriented toward system and optimization design.

### 3.3. Full Algorithm

The complete pseudocode is shown in Algorithm 1. We present only the high-level procedure here. We instantiate $g_t^\dagger$ as the median element of the benign update set $\Gamma$ for round $t$, and use $d(\cdot, \Gamma)$ as a simple shape/direction deviation metric for detecting anomalous update geometry. In each round $t$,

**Algorithm 1** Latent Optimization Attack (client $i^\star$ in round $t$)

**Require:** Global model $w_t$, real shard $\mathcal{D}_r^{(i^\star)}$, decoder $\text{Dec}(\cdot)$, latent steps $B_z$, synthetic batch size $B_s$, stepsizes $(\eta_z, \eta_w)$, reference gradient $g_t^\dagger$, budgets $(C_{\max}, \epsilon, \kappa)$

1: **Warm start:**
2: **if** $t = 1$ **then**
3:      Sample $z \sim \mathcal{N}(0, I_d)$
4: **else**
5:      Load cached $z$ from round $t - 1$
6: **end if**
7: **Latent refinement:**
8: **for** $s = 1$ **to** $B_z$ **do**
9:      $(\tilde{\boldsymbol{x}}, \tilde{y}) \leftarrow \text{Dec}(z)$
10:      $\tilde{y} \leftarrow \text{SELECTTARGETS}(\mathcal{D}_r^{(i^\star)})$
11:      $g \leftarrow \nabla_w \ell(f_{w_t}, \tilde{\boldsymbol{x}}, \tilde{y})$
12:      $\mathcal{L}_1(z) \leftarrow 1 - \frac{\langle g, g_t^\dagger \rangle}{\|g\| \|g_t^\dagger\|}$
13:      $\mathcal{L}_2(z) \leftarrow \left| \|g\| - \|g_t^\dagger\| \right|$
14:      $\mathcal{L}_3(z) \leftarrow \text{CE}(f_{w_t}(\tilde{\boldsymbol{x}}), \tilde{y})$
15:      $\mathcal{L}(z) \leftarrow \mathcal{L}_1(z) + \mathcal{L}_2(z) + \mathcal{L}_3(z)$
16:      $z \leftarrow z - \eta_z \nabla_z \mathcal{L}(z)$
17: **end for**
18: **Hybrid local training:**
19: $\tilde{\mathcal{D}} \leftarrow \{(\tilde{\boldsymbol{x}}_j, \tilde{y}_j)\}_{j=1}^{B_s}$
20: Train on $\mathcal{D}_r^{(i^\star)} \cup \tilde{\mathcal{D}}$ with stepsize $\eta_w$ to obtain $\hat{g}_t^{i^\star}$
21: **Feasibility checks:**
22: **if** $\|\hat{g}_t^{i^\star}\| > \kappa$ **then**
23:      $\hat{g}_t^{i^\star} \leftarrow \kappa \, \hat{g}_t^{i^\star} / \|\hat{g}_t^{i^\star}\|$
24: **end if**
25: **if** $d(\hat{g}_t^{i^\star}, \Gamma) > \epsilon$ **then**
26:      $\hat{g}_t^{i^\star} \leftarrow \text{PROJECTTOBENIGN}(\hat{g}_t^{i^\star})$
27: **end if**
28: Ensure $C(\hat{g}_t^{i^\star}) \leq C_{\max}$
29: **Report & cache:**
30: Upload $\hat{g}_t^{i^\star}$ and cache $(z, w_t)$ for round $t + 1$

the attacker warm-starts $z$, performs up to $B_z$ latent steps on the joint objective $\mathcal{L}(z)$ (see Section 3.2.2), decodes a compact batch, mixes it with real data for local training, and reports $\hat{g}_t^{i^\star}$. Before reporting, we enforce norm and deviation constraints $(\epsilon, \kappa)$ and a communication budget $C_{\max}$; if any constraint is violated, we project $\hat{g}_t^{i^\star}$ to a benign subspace or revert to a benign update. This feasibility check keeps runtime and communication comparable to standard FL while ensuring that attribution gains persist without producing visibly anomalous updates.

### 3.4. Theoretical Analysis

We briefly explain why the proposed latent optimization attack preserves global utility while inflating client-level

attribution scores. For Shapley-style evaluators such as FedSV, the per-round marginal utility is defined as $\Delta_t(g) = U(w_t + g) - U(w_t)$, and the final attribution aggregates these gains across rounds and permutations. Consequently, even small but persistent utility improvements can accumulate and substantially influence the final attribution score.

Let $w_t$ denote the global model at round $t$. If client $i^\star$ behaves benignly, local training on its real dataset $\mathcal{D}_r^{(i^\star)}$ produces a standard update $g_t^{(r)}$. Under latent optimization, the client uploads the following hybrid update:

$$\hat{g}_t^{i^\star} = (1 - \alpha) \, g_t^{(r)} + \alpha \, g_t^{(s)} \tag{8}$$

where $g_t^{(s)}$ is induced by latent-optimized synthetic samples and $0 < \alpha \ll 1$ controls the synthetic contribution.

The latent optimization objective constrains the synthetic component to align with the global descent direction while maintaining a magnitude comparable to benign updates. Under standard smoothness assumptions, this alignment yields a non-negative marginal utility rewarded by FedSV, while the small coefficient $\alpha$ keeps the hybrid update dominated by the benign training signal and avoids large deviations from benign optimization behavior. Consequently, the synthetic component influences attribution evaluation without substantially perturbing the optimization trajectory. As a result, the attack preserves standard training dynamics and causes negligible degradation in global utility across rounds.

Under non-IID partitioning, latent optimization targets missing or under-represented labels when generating synthetic samples. Although the synthetic component is small, the resulting coverage-driven utility gains accumulate across rounds and client subsets in marginal-utility evaluators such as FedSV (Wang et al., 2020), leading to inflated attribution scores for client $i^\star$. Furthermore, the hybrid structure keeps $\hat{g}_t^{i^\star}$ statistically close to benign updates, allowing the attack to evade utility-centric and geometry-based defenses while remaining consistent with standard aggregation dynamics.

## 4. Experimental Evaluation

### 4.1. Experimental Setup

**Datasets and models.** We evaluate on three image benchmarks—CIFAR-10 (Krizhevsky et al., 2009), SVHN (Netzer et al., 2011), and FashionMNIST (Xiao et al., 2017) —using standard train/test splits. We report results across three backbones: ResNet-18 (He et al., 2016), WRN-28-10 (Zagoruyko & Komodakis, 2016), and VGG16_BN (Simonyan & Zisserman, 2014).

**Partitioning and FL protocol.** Unless otherwise stated, we use a standard FedAvg setup with $N=10$ clients. Training data are partitioned in a non-IID class-imbalanced manner so that each client observes a subset of classes with roughly 3,000 samples (Zhao et al., 2018); an IID split is used only

for sanity checks. In each communication round $t$, the server broadcasts $w_t$, clients train locally, return their updates, and the server aggregates a data-size–weighted average to obtain $w_{t+1}$ (McMahan et al., 2017).

**Evaluators and metrics.** Our primary evaluator is Federated Shapley Value (Wang et al., 2020); we also report Leave-One-Out (LOO) (Vehtari et al., 2017) for robustness. We track each client's data attribution value and its rank among peers (before/after attack), and we report test accuracy to verify utility preservation.

**Normalization of attribution values.** Raw attribution values produced by FedSV and LOO can be positive or negative. For cross-client comparison and rank-based analysis, we therefore report a normalized attribution share obtained via a shift-based linear normalization that subtracts the minimum attribution value and rescales all client contributions to sum to one. Unless otherwise stated, figures and rankings are based on these normalized shares.

**Attacks compared.** We compare five settings: Attack-Free (no manipulation), Label-Flip (Xiao et al., 2012), Random-Noise (Shahani & Scheutz, 2025), Free-Rider (Fraboni et al., 2021), and our Latent Optimization attack. Unless noted, one malicious client is used (the lowest-attribution client in the attack-free run).

**Latent Optimization details.** The attacker holds a fixed pre-trained decoder trained on the same image domain as the downstream task, and in each round, optimizes a low-dimensional latent vector against the broadcast global model to generate a compact synthetic batch. Synthetic samples are mixed with real data during local training, with labels biased toward underrepresented classes to exploit evaluator sensitivity to coverage.

### 4.2. Experiment Results

**Main Attribution Gains under FedSV.**

Table 1 reports the malicious client's attribution value and rank across datasets and architectures. In the attack-free setting, the malicious client consistently receives the lowest attribution and ranks last, indicating that FedSV identifies it as a negligible contributor with low marginal utility. Under Latent Optimization, attribution scores increase substantially across settings, moving the malicious client into the upper half of the ranking and, in several cases, close to highly contributing benign participants. In contrast, baseline attacks such as Label Flipping, Random Noise, and Free Rider produce negligible or inconsistent shifts and never achieve high attribution. Thus, Latent Optimization uniquely reshapes attribution outcomes under identical training protocols, outperforming conventional adversarial attacks in attribution manipulation.

*Table 1.* Comparison of the Malicious Client's Normalized Data Attribution Value and Rank Before and After the Attack Under FedSV.

*(a)* CIFAR-10

| Attack Method | Data Attribution Value | | | Data Attribution Rank | | |
|---|---|---|---|---|---|---|
| | ResNet-18 | WRN-28-10 | VGG16_BN | ResNet-18 | WRN-28-10 | VGG16_BN |
| Attack Free | 0.0547 | 0.0001 | 0.0000 | 10 | 10 | 10 |
| Label Flipping | 0.0591 | 0.0001 | 0.0454 | 8 | 10 | 8 |
| Random Noise | 0.0074 | 0.0001 | 0.0496 | 10 | 10 | 8 |
| Free Rider | 0.0802 | 0.0001 | 0.0468 | 8 | 10 | 9 |
| Latent Optimization | 0.1682 | 0.0761 | 0.1160 | 2 | 6 | 4 |

*(b)* FashionMNIST

| Attack Method | Data Attribution Value | | | Data Attribution Rank | | |
|---|---|---|---|---|---|---|
| | ResNet-18 | WRN-28-10 | VGG16_BN | ResNet-18 | WRN-28-10 | VGG16_BN |
| Attack Free | 0.0000 | 0.0000 | 0.0000 | 10 | 10 | 10 |
| Label Flipping | 0.0338 | 0.0143 | 0.0382 | 7 | 9 | 9 |
| Random Noise | 0.0269 | 0.0286 | 0.0368 | 8 | 9 | 9 |
| Free Rider | 0.0198 | 0.0264 | 0.0398 | 8 | 9 | 9 |
| Latent Optimization | 0.0741 | 0.0891 | 0.1072 | 6 | 7 | 5 |

*(c)* SVHN

| Attack Method | Data Attribution Value | | | Data Attribution Rank | | |
|---|---|---|---|---|---|---|
| | ResNet-18 | WRN-28-10 | VGG16_BN | ResNet-18 | WRN-28-10 | VGG16_BN |
| Attack Free | 0.0000 | 0.0000 | 0.0000 | 10 | 10 | 10 |
| Label Flipping | 0.0000 | 0.0232 | 0.0257 | 10 | 8 | 9 |
| Random Noise | 0.0096 | 0.0337 | 0.0300 | 9 | 7 | 9 |
| Free Rider | 0.0000 | 0.0000 | 0.0023 | 10 | 10 | 9 |
| Latent Optimization | 0.0933 | 0.0947 | 0.1256 | 6 | 7 | 6 |

Beyond absolute gains, Latent Optimization perturbs the attribution structure among benign clients. As shown in Appendix Figures 5–13, attribution mass is redistributed across benign participants, altering rankings and share proportions. Because FedSV aggregates marginal utilities over permutations, even small coverage and alignment advantages accumulate across rounds and are amplified by the evaluator. Consequently, the attack not only inflates the malicious client's attribution, but also reshapes relative credit assignment among benign participants. These results show that attribution can be manipulated without degrading global utility and that FedSV is not robust to incentive-aligned attacks targeting the evaluator rather than model accuracy.

**Utility Preservation under FedSV.** Table 2 summarizes global model accuracy under different attacks. Label Flipping and Random Noise consistently degrade test performance, reflecting perturbations to the decision boundary or optimization dynamics during collaborative training. Free Rider largely preserves utility but does not improve attribution, indicating that avoiding harmful updates alone is insufficient for attribution inflation. In contrast, Latent Optimization maintains accuracy within a narrow margin of the attack-free baseline across datasets and architectures while consistently increasing the attacker's measured contribution.

These results show that attribution manipulation does not require harming model utility. By aligning with global descent directions and preserving task fidelity, the attacker can systematically boost its measured contribution while remaining compatible with standard training dynamics and aggregation. This contrasts with traditional poisoning attacks, which of-

*Table 2.* Comparison of Global Model Accuracy Before and After Malicious Client Attacks Under FedSV.

*(a)* CIFAR-10

| Attack Method | Models | | |
|---|---|---|---|
| | ResNet-18 | WRN-28-10 | VGG16_BN |
| Attack Free | 76.75% | 70.12% | 71.57% |
| Label Flipping | 64.68% | 61.21% | 62.47% |
| Random Noise | 66.28% | 63.58% | 64.96% |
| Free Rider | 70.63% | 61.98% | 67.54% |
| Latent Optimization | 77.59% | 77.23% | 69.66% |

*(b)* FashionMNIST

| Attack Method | Models | | |
|---|---|---|---|
| | ResNet-18 | WRN-28-10 | VGG16_BN |
| Attack Free | 73.11% | 64.45% | 73.19% |
| Label Flipping | 60.36% | 56.62% | 66.32% |
| Random Noise | 61.23% | 56.34% | 68.33% |
| Free Rider | 71.38% | 65.53% | 71.41% |
| Latent Optimization | 72.23% | 60.07% | 69.62% |

*(c)* SVHN

| Attack Method | Models | | |
|---|---|---|---|
| | ResNet-18 | WRN-28-10 | VGG16_BN |
| Attack Free | 84.11% | 75.67% | 83.52% |
| Label Flipping | 70.83% | 64.10% | 72.37% |
| Random Noise | 67.97% | 67.70% | 76.39% |
| Free Rider | 82.14% | 76.13% | 83.29% |
| Latent Optimization | 81.20% | 77.74% | 81.13% |

ten trade utility for adversarial influence, and further shows that performance-based defenses are inadequate for protecting attribution integrity under marginal-utility evaluators.

**Comparison with Direct Reference Alignment.** To isolate the role of latent optimization beyond gradient alignment, we compare our method with a direct reference-direction baseline. In this baseline, the malicious client aligns its update with a benign reference gradient and rescales it to match the norm of a benign update, preserving approximate optimization magnitude and overall update scale. In contrast, our method generates synthetic samples through latent optimization and performs local training on mixed real and synthetic batches, allowing the attacker to optimize both gradient direction and task utility simultaneously.

Table 3 shows that direct reference alignment alone can partially increase attribution value and improve attribution rank, indicating that alignment with the global optimization trajectory already affects marginal-utility evaluators. However, our method consistently achieves larger attribution gains across evaluated targets while maintaining comparable utility and stable training behavior. These results suggest that directional alignment alone is insufficient to fully exploit the attribution mechanism underlying FedSV. Latent optimization additionally enforces task fidelity and magnitude consistency through supervised training on synthetic samples, producing updates that are both optimization-aligned and effective under the learning objective. As a result, the

*Table 3.* Comparison of the direct reference-direction baseline and the full method.

| Target Rank | Method | Data Attribution Value | Data Attribution Rank | Global Model Accuracy (%) |
|---|---|---|---|---|
| Rank 2 | Direct Ref | 0.1243 | 2 | 72.21 |
| | Full Method | **0.1632** | **1** | 73.59 |
| Rank 6 | Direct Ref | 0.0871 | 5 | 73.88 |
| | Full Method | **0.1011** | **4** | 73.94 |
| Rank 10 | Direct Ref | 0.1053 | 7 | 72.63 |
| | Full Method | **0.1627** | **4** | 75.39 |

generated updates receive substantially higher attribution under FedSV than updates produced through direction alignment, demonstrating that attribution inflation depends not only on alignment direction but also on utility-compatible optimization structure and training-induced gradient consistency.

**Comparison with Centralized Attribution Attacks.** We compare the proposed method with two representative attribution manipulation attacks adapted from centralized learning: Shadow Attack and Outlier Attack. Although both methods effectively inflate data attribution in centralized settings, their effectiveness does not directly transfer to federated learning, where attribution depends on aggregated client updates rather than individual data points and therefore reflects collaborative optimization behavior. As shown in Table 4, both baselines achieve limited or inconsistent attribution gains across evaluated targets. In contrast, Latent Optimization consistently achieves higher attribution values and stronger attribution ranks while maintaining higher global accuracy and stable optimization behavior.

These results suggest that attribution manipulation in federated learning fundamentally differs from centralized data-level attribution attacks. Federated attribution methods evaluate client updates through marginal contribution to global utility rather than individual data attribution. Consequently, attacks based on high-attribution samples or adversarial outliers do not reliably translate into high-utility client updates after local training and aggregation, since their attribution signals can be diluted or misaligned during collaborative optimization. In contrast, Latent Optimization directly optimizes utility-aligned client updates through latent-generated synthetic batches, systematically inflating attribution while remaining compatible with standard training dynamics and federated aggregation across communication rounds.

*Table 4.* Comparison between Latent Optimization and existing attribution manipulation attacks adapted to federated learning.

| Benign Target Rank | Method | Attribution Value | Attribution Rank | Accuracy (%) |
|---|---|---|---|---|
| Rank 2 | Shadow | 0.1248 | 3 | 71.92 |
| | Outlier | 0.0916 | 5 | 67.47 |
| | Latent Optimization | **0.1522** | **1** | **73.59** |
| Rank 6 | Shadow | 0.0817 | 5 | 73.88 |
| | Outlier | 0.0664 | 7 | 69.12 |
| | Latent Optimization | **0.1063** | **4** | **76.94** |
| Rank 10 | Shadow | 0.1129 | 6 | 72.75 |
| | Outlier | 0.0798 | 8 | 68.85 |
| | Latent Optimization | **0.1547** | **4** | **74.39** |

**Scaling with the Number of Clients.** Figure 2 varies the total number of clients across $\{7, 10, 12, 15\}$ and reports the malicious client's attribution (ranked last in the attack-free baseline) under different attacks. Under the attack-free baseline, the malicious client consistently receives near-zero attribution across all scales, indicating that the evaluator identifies it as a negligible contributor with limited marginal utility under standard collaborative optimization. In contrast, Latent Optimization yields substantially higher attribution for all evaluated client counts, peaking at 10 clients and decreasing slightly at 12 and 15. Meanwhile, Label Flipping, Random Noise, and Free Rider remain near zero across settings, indicating that they do not benefit from larger client populations or complex aggregation structure under marginal-utility evaluation.

These trends show that latent optimization scales to larger client cohorts and is not limited to small participation settings. The slight drop at higher client counts reflects attribution normalization across more participants rather than attack failure, since the malicious client still maintains a substantial relative attribution advantage throughout training and aggregation. Crucially, the attacker preserves this advantage by exploiting non-IID label coverage and evaluator structure, while competing attacks lack such alignment and utility compatibility. Overall, the results suggest that the vulnerability stems from the marginal-utility attribution mechanism and persists under realistic deployment scales and heterogeneous participation regimes with varying client populations.

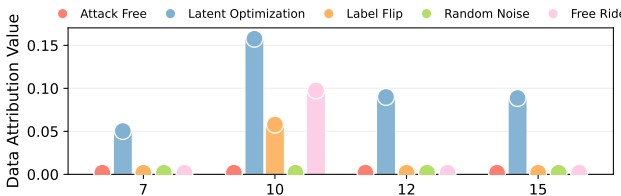

*Figure 2.* Grouped bar plot of data attribution values across attack methods under different client counts.

**Effect of Malicious-Client Selection.** Figure 3 varies the malicious participant by selecting the client with attack-free FedSV rank $2, 4, 6, 8,$ or $10$. When the attacker starts from a high-rank position, Latent Optimization does not substantially increase attribution; scores remain near baseline or slightly decrease, yet still outperform competing attacks, which heavily penalize the attacker once training is perturbed and optimization alignment deteriorates. When the attacker starts from a low-rank position, Latent Optimization produces substantial attribution uplift, elevating previously under-valued clients into mid-to-high ranges, while competing attacks further suppress contribution and marginal utility under evaluator aggregation.

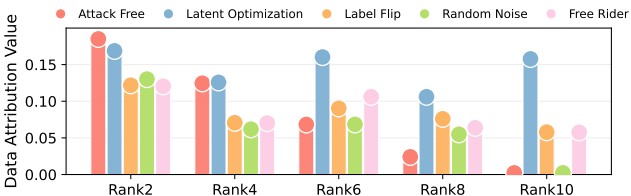

*Figure 3.* **Effect of malicious-client selection.** Grouped bar plot (with colored dots marking each value, including near-zero cases) of client-level data attribution values across different selection strategies.

These results indicate that latent optimization benefits both strong and weak contributors, but with asymmetric gains. High-ranking clients are already near evaluator saturation and remain stable, whereas low-ranking clients retain substantial "headroom" to convert missing label coverage into marginal utility through optimization-aligned synthetic updates and coverage-aware utility gains. Notably, this uplift occurs without harming global performance, allowing FedSV to reassign credit toward these initially small contributors across communication rounds. This pattern is absent in baseline attacks and highlights that attribution manipulation is especially effective for clients initially appearing benign, weakly contributing, and statistically insignificant under standard attribution evaluation and utility-based ranking.

**Robustness Across Evaluators.** To assess whether attribution manipulation depends on a specific evaluator, we repeat the experiments using Leave-One-Out (LOO), which measures marginal utility by removing one client and quantifying its effect on global performance and aggregated model utility. Figure 15 shows a representative dataset-architecture pair. Under the attack-free baseline, the malicious client again occupies the lowest attribution region, indicating that LOO also identifies it as a negligible contributor under standard collaborative training. In contrast, under Latent Optimization, attribution mass shifts toward the malicious client and improves its rank among peers. Competing attacks produce only weak or inconsistent shifts. Meanwhile, global accuracy remains comparable across settings (Figure 16), indicating that the attribution changes do not rely on degrading utility or destabilizing optimization.

Taken together, the results show that attribution inflation from latent optimization generalizes beyond FedSV. Although LOO and FedSV differ in mechanism and estimator variance, both assign substantially higher attribution to the malicious client and perturb attribution structure among benign clients similarly and consistently across evaluation settings. This suggests that the vulnerability stems from the incentive properties of marginal-utility evaluation rather than any specific scoring rule or estimator. To avoid redundancy across evaluators and settings, we report only one appendix configuration, all exhibiting consistent trends and utility-preserving behavior throughout training.

**Effect of Attack Intensity.** Figure 4 varies attack intensity from $0\times$ to $4\times$ and reports attribution values and test accuracy. Attribution increases monotonically with higher intensity: even low-intensity synthetic coverage ($0.25\times$ and $0.50\times$) raises the malicious client's estimated contribution, while stronger intensities ($2\times$ and $4\times$) further amplify attribution gain. Notably, the increase remains smooth across intensity levels, indicating that attribution inflation scales predictably and consistently with synthetic coverage strength. In contrast, test accuracy remains within a narrow range and does not degrade with increasing intensity; stronger attacks do not induce utility drops and cause only minor fluctuations around baseline accuracy.

These results show that latent optimization supports controllable scaling: increasing attack strength systematically boosts the attack objective, marginal contribution, without triggering utility deterioration that typically exposes poisoning behaviors. Furthermore, stable test accuracy suggests that the attack operates by enhancing the evaluator's notion of beneficial coverage rather than harming global convergence or optimization dynamics during collaborative training. This decoupling between attribution inflation and utility preservation highlights a key vulnerability of marginal-utility attribution: an attacker can scale its influence without trade-offs seen in performance-driven attacks or being flagged by accuracy-based defenses.

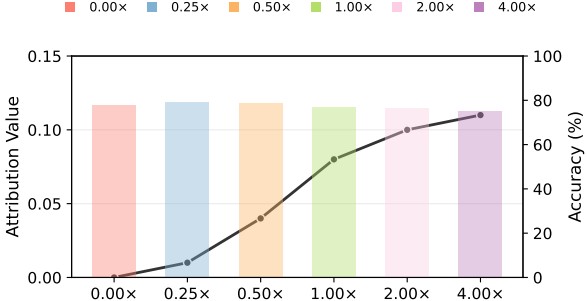

*Figure 4.* **Effect of attack intensity on data attribution.** Normalized attribution share and test accuracy under increasing attack intensity.

## 5. Defenses

There is currently no defense specifically designed to protect client-level data attribution in FL. To assess whether standard utility-centric defenses can incidentally mitigate attribution manipulation, we evaluate a widely used baseline: geometry-based trimming (Steinhardt et al., 2017), which filters out updates that deviate from the bulk of client updates and is commonly deployed to defend against data poisoning and anomalous optimization behavior.

We apply geometry-based trimming at the server in each

*Table 5.* **Per-round detection performance against the Latent Optimization attack.** Precision, Recall, and F1-score are averaged across communication rounds.

| Method / Setting | Precision | Recall | F1-Score |
|---|---|---|---|
| Random Guess (1 of 10) | 0.10 | 0.10 | 0.10 |
| Geometry-based Trimming | | | |
|     CIFAR-10 + ResNet-18 | 0.00 | 0.00 | 0.00 |
|     FashionMNIST + WRN-28-10 | 0.02 | 0.01 | 0.01 |
|     SVHN + VGG16_BN | 0.01 | 0.02 | 0.01 |

round and interpret the robust aggregation rule as an implicit detector: a client is regarded as detected in a round if its update is trimmed or excluded. We formulate defense effectiveness as a per-round malicious-client detection task and report Precision, Recall, and F1-score averaged across communication rounds. A random-guess baseline (selecting one of $N{=}10$ clients per round) is included for reference and to contextualize the detection difficulty.

Table 5 summarizes the detection performance across three dataset–model combinations. Across all settings, geometry-based trimming fails to reliably identify the latent optimization attacker: Precision and Recall remain at or near zero, and F1-scores are comparable to or worse than random guessing. This behavior is expected: as shown in Section 3.4, latent optimization produces hybrid updates that remain aligned with the global descent direction and statistically embedded within the benign update distribution, thereby evading geometry-based defenses and remaining difficult to distinguish from benign optimization behavior.

## 6. Conclusion

We show that data attribution in distributed learning can be strategically manipulated without harming global utility. A single participant, through latent optimization on a latent vector, consistently inflates its attribution under data attribution evaluators. The attack redistributes attribution among benign clients, scales across datasets and models, and evades standard utility- and geometry-based defenses. These results reveal a new attack surface in data attribution mechanisms, motivating future work on attribution-robust evaluators and incentive designs. More broadly, our findings identify attribution integrity as an important security consideration in distributed learning systems.

**Limitations and Outlook.** Our analysis focuses on a single attacker and marginal-utility evaluators. Extending the study to broader settings, including dynamic participation, multiple adversaries, and more diverse incentive mechanisms, remains important future work. Investigating defenses specifically designed for attribution robustness is another promising direction.

## Acknowledgements

We thank our collaborators and colleagues for helpful discussions and feedback throughout this work. We also thank the anonymous reviewers for their valuable feedback and constructive suggestions. This work used computational resources provided by Auburn University.

## Impact Statement

This work examines the integrity of data attribution mechanisms and shows that attribution can be manipulated without harming global utility. While such manipulation could be misused to unfairly obtain credit in distributed learning systems, our goal is to reveal this overlooked vulnerability and thereby inform the design of attribution-robust evaluators, auditing tools, and incentive mechanisms. As attribution metrics increasingly govern compensation, provenance, and participation rights in ML ecosystems, improving attribution integrity has a positive influence on the development of trustworthy ML infrastructure.

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

# A. Background and Related Work

## A.1. Federated Learning

Federated learning (FL) enables collaborative training over decentralized data without sharing raw samples. Classical FL follows the broadcast–local-train–aggregate loop, where a central server coordinates many clients under privacy, communication, and heterogeneity constraints (Liu et al., 2024; Papadopoulos et al., 2024; Liu et al., 2025; Murad et al., 2025). Modern surveys highlight FL's defining characteristics: naturally non-IID and unbalanced data partitions, limited client availability, partial participation, and system-level constraints such as communication budgets and straggler resilience (Yurdem et al., 2024; Kaur et al., 2024). These properties make FL attractive for IoT analytics, healthcare, autonomous vehicles, and smart city infrastructure (Schoinas et al., 2024), but also widen the gap between theory and practice, creating structural vulnerabilities that adversaries can exploit.

A recurring challenge is non-IID data, where label distribution skews across clients degrade convergence and destabilize aggregation (Wang et al., 2021). Recent work on FL robustness and trustworthiness focuses on secure aggregation, incentive-aligned participation, and model-centric defenses (Lyu et al., 2020; Liu et al., 2022; Li et al., 2025), but they often assume that participants aim to preserve accuracy rather than manipulate attribution, leaving attribution-specific vulnerabilities insufficiently addressed.

## A.2. Training Data Attribution

Training data attribution seeks to quantify the influence of individual samples or subsets on model behavior. Foundational frameworks formalize three components: (i) the model behavior to be explained, (ii) the training entities to be credited, and (iii) the influence measure used to score them (Zhang et al., 2025b). Recent surveys show that attribution underpins data governance, auditing, pricing, and debugging in modern Machine Learning pipelines (Deng et al., 2025; Hammoudeh & Lowd, 2024).

Four major families of attribution methods have emerged:

(1) **Influence-function methods.** These use second-order approximations of leave-one-out retraining to estimate a sample's contribution. Advances tackle computational bottlenecks and improve scalability (Ramu et al., 2024; Sun et al., 2024a; Zheng et al., 2023; Kwon et al., 2023; Mlodozeniec et al., 2024; Lin et al., 2024b). They are widely used in debugging and understanding model vulnerabilities.

(2) **Marginal-contribution methods.** Shapley- and Banzhaf-style valuations treat training as a cooperative game, estimating expected contributions across subsets. Extensions address robustness, fragmentation, privacy, and efficiency (Chen et al., 2025; Gairola et al., 2025; Li & Yu, 2023; Wang et al., 2024c; Sun et al., 2024b; Garrido Lucero et al., 2024; Covert et al., 2024b). Opendataval benchmarks highlight the difficulty of evaluating attribution across tasks and models (Jiang et al., 2023; Jiao et al., 2025).

(3) **Training-dynamics approaches.** These methods monitor intermediate checkpoints or trajectories to attribute influence over the course of training (Bae et al., 2024; Wei et al., 2024; Wang et al., 2024a; Xu et al., 2024; Hu et al., 2025; Sun et al., 2025). Such approaches capture nuanced temporal effects and are increasingly used to study fine-tuning and generative models.

(4) **Simulator- and surrogate-based estimators.** These methods approximate counterfactual outcomes using auxiliary models or sampling-based surrogates, enabling efficient estimation in large-scale pipelines (Covert et al., 2024a; Sun et al., 2024a; Schioppa, 2024; Mlodozeniec et al., 2026; Lin et al., 2024a). Attribution has expanded to diffusion and text-to-image systems (Wang et al., 2024d; Lin et al., 2024b).

Despite technical progress, the literature rarely considers adversarial participants who strategically manipulate attribution scores while maintaining high utility. Only very recent work begins examining explicit threats to attribution (Wang et al., 2024e), but these analyses do not explore latent-space optimization nor hybrid real–synthetic manipulation as studied in this work.

## A.3. Data Poisoning and Adversarial Manipulation

Data poisoning attacks modify or inject malicious samples to subvert training. Classical poisoning includes label-flipping, feature perturbations, and generative-model–based sample synthesis, which can be highly covert in distributed environments (Sardana et al., 2024; Nowroozi et al., 2025; Xiao et al., 2024). Deep networks' capacity to memorize small poisoned subsets makes subtle manipulations difficult to detect (Zhao et al., 2025). Recent work shows that poisoning can persist even under model unlearning or retraining (Pawelczyk et al., 2024).

In federated settings, poisoning is more challenging due to decentralized control. Studies explore client-level malicious behavior, including Byzantine updates, targeted poisoning, and defenses based on robust aggregation or anomaly detection (Jia et al., 2024; Bal et al., 2025). However, these defenses largely assume that adversaries aim to degrade performance. Our work targets a complementary and underexplored setting: adversaries who preserve global utility while manipulating contribution metrics.

Parallel to poisoning, valuation-based attacks have begun to emerge. Wang et al. (Wang et al., 2024e) studied manipulations of attribution under simplified conditions, but do not

examine latent-space optimization, cross-round consistency, or FL-specific non-IID structure. Our latent-optimization attack fills this gap by demonstrating that attribution can be steered consistently, stealthily, and without harming accuracy.

## B. Notation

| Symbol | Description |
|---|---|
| $\mathcal{X}, \mathcal{Y}$ | Input and label spaces of the supervised learning task. |
| $N$ | Number of clients (in this paper, $N{=}10$). |
| $t$ | Communication-round index. |
| $T$ | Total number of federated learning rounds. |
| $w_t$ | Global model parameters broadcast by the server at round $t$. |
| $f_{w_t}$ | Model instantiated with parameters $w_t$. |
| $U(\cdot)$ | Global utility metric (e.g., test accuracy). |
| $U^{(0)}$ | Final utility of the clean (no-attack) training run. |
| $U^{(1)}$ | Final utility of the training run under attack. |
| $\delta$ | Allowed tolerance on utility degradation, cf. Eq. (1). |
| $i^\star$ | Index of the malicious client. |
| $\mathcal{D}_r^{(i^\star)}$ | Real (benign) local dataset of client $i^\star$. |
| $\mathcal{D}_s$ | Synthetic dataset constructed by the attacker (must lie in $\mathcal{X} \times \mathcal{Y}$). |
| $\tilde{\mathcal{D}}$ | Synthetic batch decoded from the latent vector $z$ for local training. |
| $(\tilde{\boldsymbol{x}}, \tilde{y})$ | Synthetic input–label pair decoded from $z$. |
| $g_t^{i^\star}$ | Benign local update of client $i^\star$ at round $t$. |
| $\hat{g}_t^{i^\star}$ | Reported (possibly malicious) update of client $i^\star$ at round $t$. |
| $g_t^{(r)}$ | Update induced by training only on $\mathcal{D}_r^{(i^\star)}$. |
| $g_t^{(s)}$ | Update induced by training on the synthetic batch only. |
| $\alpha$ | Synthetic weight in the hybrid update, cf. Eq. (8). |
| $E(\cdot)$ | Server-side data-attribution evaluator. |
| $C(\cdot)$ | Communication cost of an update (budget constraint), cf. Eq. (3). |
| $\Gamma$ | Reference set of benign client updates used for deviation checks. |
| $d(\cdot, \Gamma)$ | Deviation metric from benign updates (shape/direction), cf. Eq. (3). |
| $C_{\max}$ | Upper bound on communication cost (budget). |
| $\epsilon$ | Deviation budget controlling how far an update may deviate from benign ones. |
| $\kappa$ | Norm budget for client updates (maximum allowed $\ell_2$ norm). |
| $z$ | Latent vector optimized by the attacker to parameterize synthetic samples. |
| $\mathrm{Dec}(\cdot)$ | Fixed decoder/generator that maps $z$ to synthetic samples. |
| $\ell(\cdot)$ | Supervised task loss used to define gradients. |
| $\mathrm{CE}(\cdot, \cdot)$ | Cross-entropy loss on model predictions and labels. |
| $\mathcal{L}_1(z)$ | Loss term enforcing directional alignment between synthetic and reference gradients. |
| $\mathcal{L}_2(z)$ | Loss term enforcing norm consistency between synthetic and reference gradients. |
| $\mathcal{L}_3(z)$ | Task loss on synthetic samples to maintain class-discriminative features. |
| $\mathcal{L}(z)$ | Joint latent objective: $\mathcal{L}_1 + \mathcal{L}_2 + \mathcal{L}_3$. |
| $g_t^\dagger$ | Reference benign gradient at round $t$ (e.g., approximated via $w_t - w_{t-1}$). |
| $\theta(z)$ | Angle between the synthetic gradient and the reference gradient at round $t$. |
| $\nabla_w \ell(f_{w_t}, \mathrm{Dec}(z))$ | Gradient of the task loss on the decoded synthetic batch at $w_t$. |
| $\Delta_t(g)$ | Per-round marginal utility of update $g$ at round $t$, i.e., $U(w_t + g) - U(w_t)$. |
| $\eta_z$ | Step size for latent optimization in $z$-space. |
| $\eta_w$ | Local learning rate used in client-side model training. |
| $B_z$ | Number of gradient steps for latent optimization in each round. |
| $B_s$ | Synthetic batch size decoded from $z$ for local training. |

## C. Experiment results

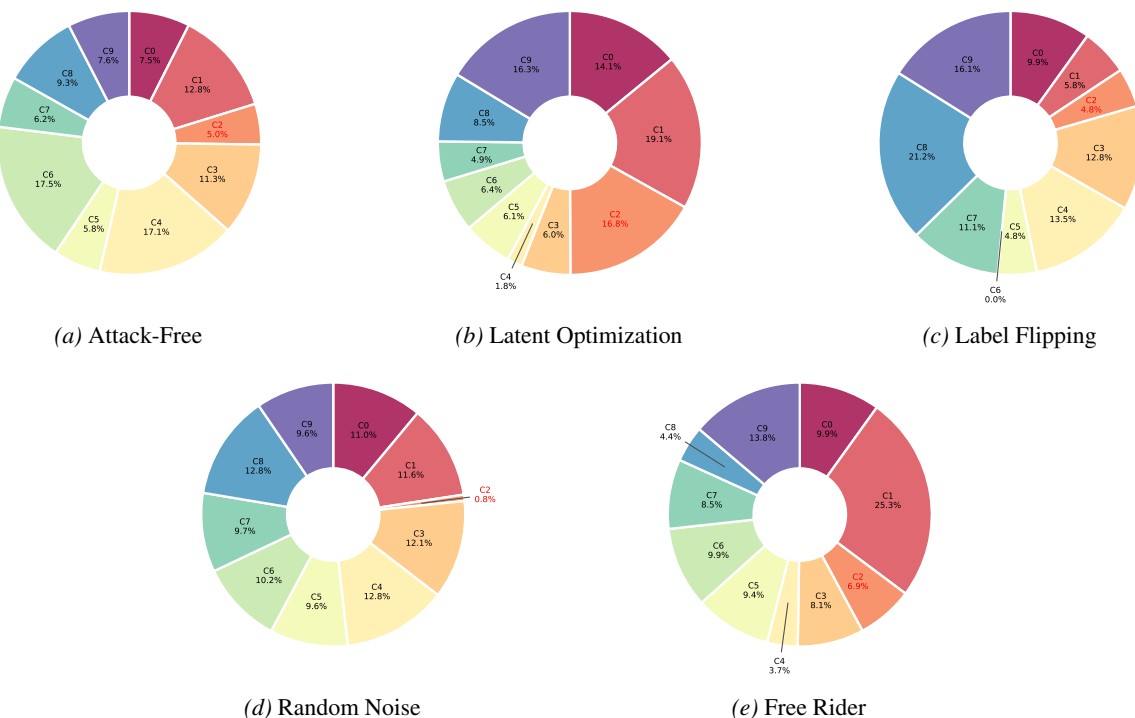

*Figure 5.* **Client-level data attribution under CIFAR-10 with ResNet-18.** Per-client attribution shares under different attack settings. Colors indicate client identities and are consistent across panels.

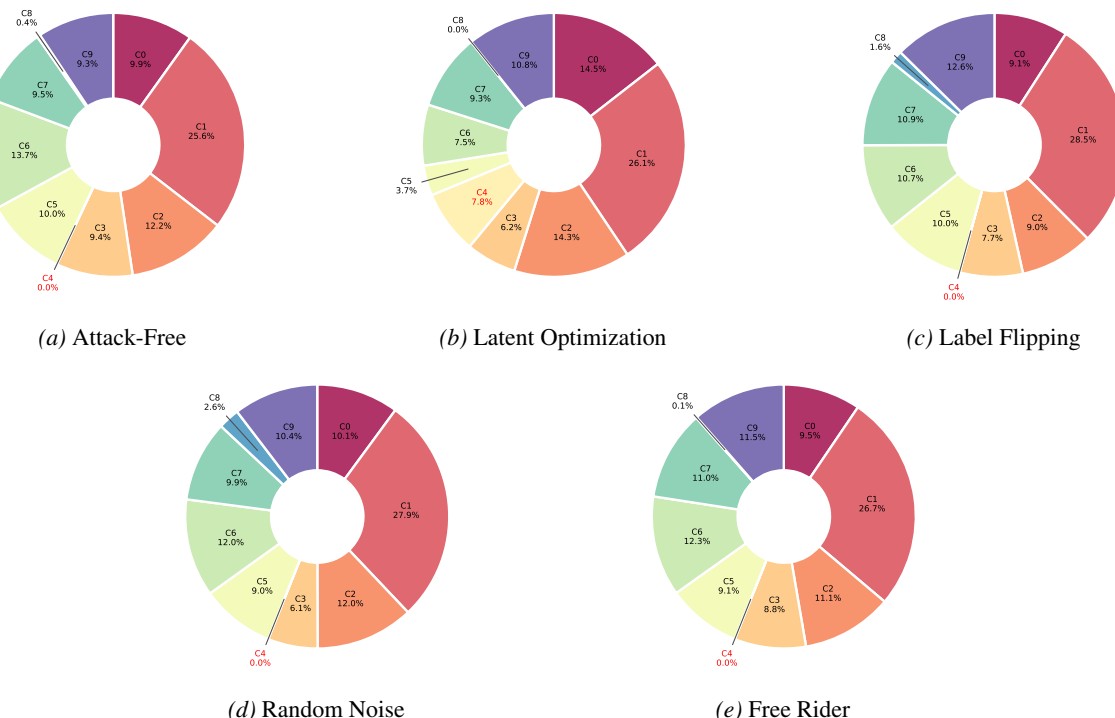

*Figure 6.* **Client-level data attribution under CIFAR-10 with WRN-28×10.** Per-client attribution shares under different attack settings. Colors indicate client identities and are consistent across panels.

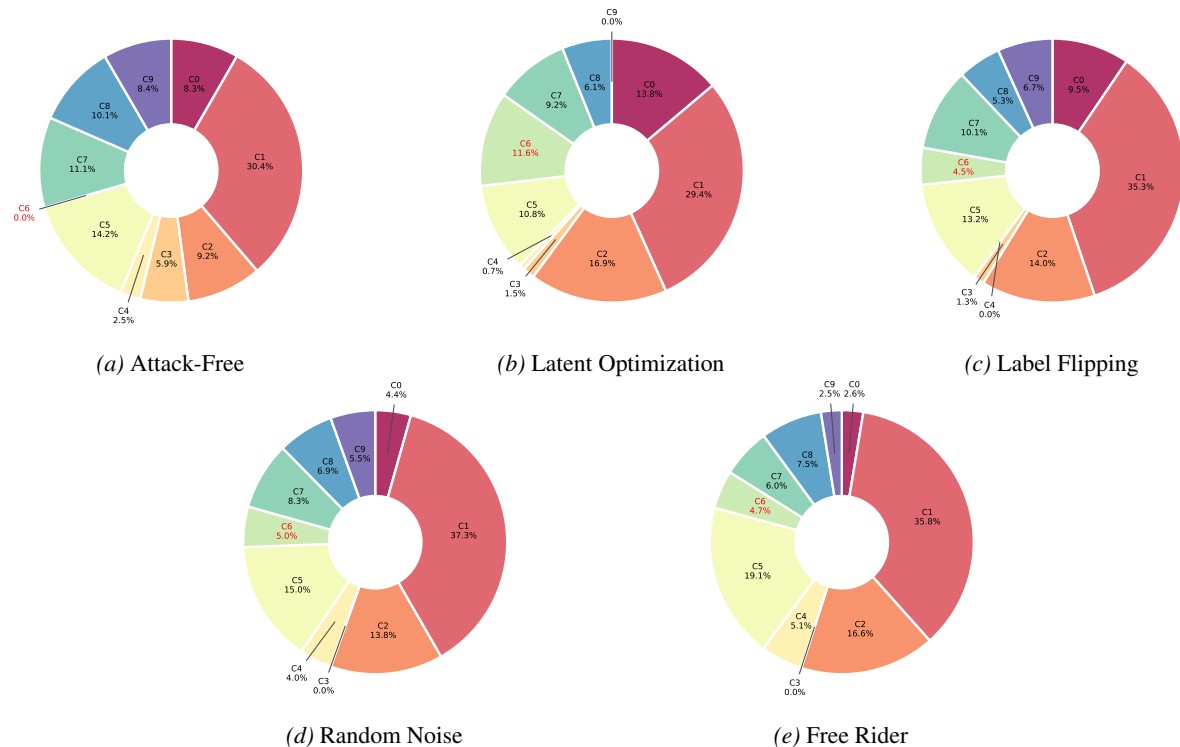

*Figure 7.* **Client-level data attribution under CIFAR-10 with VGG16_BN.** Per-client attribution shares under different attack settings. Colors indicate client identities and are consistent across panels.

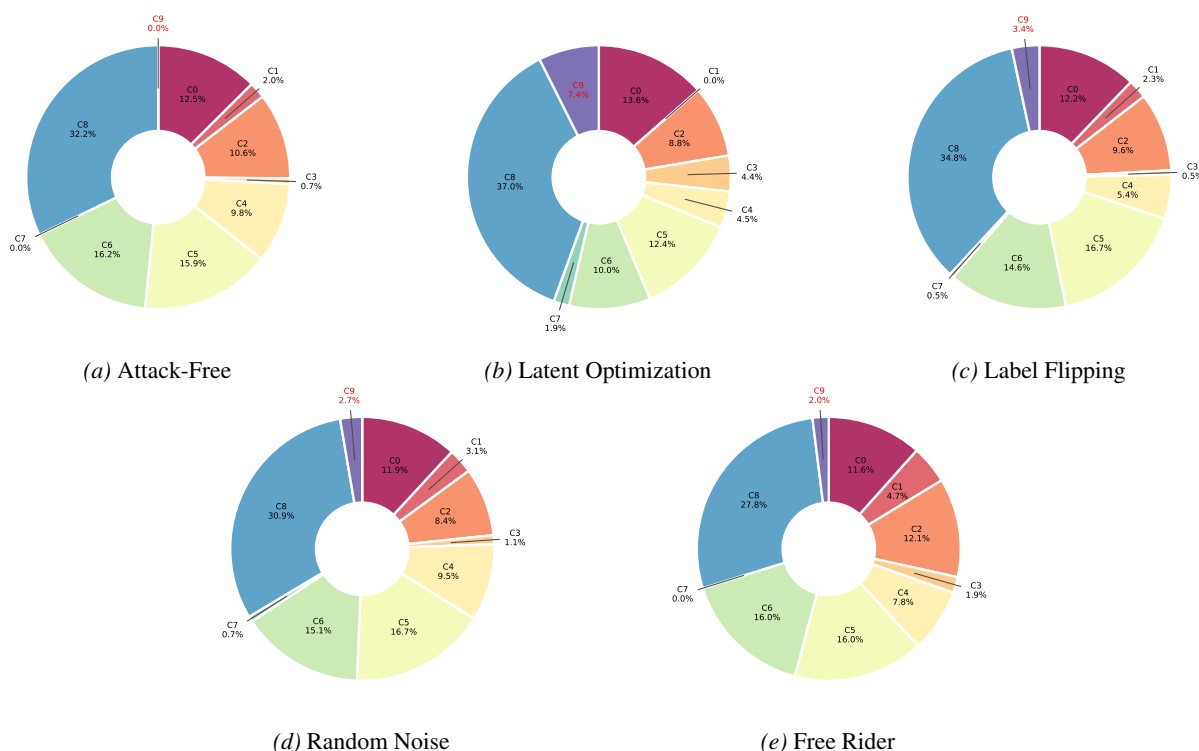

*Figure 8.* **Client-level data attribution under FashionMNIST with ResNet-18.** Per-client attribution shares under different attack settings. Colors indicate client identities and are consistent across panels.

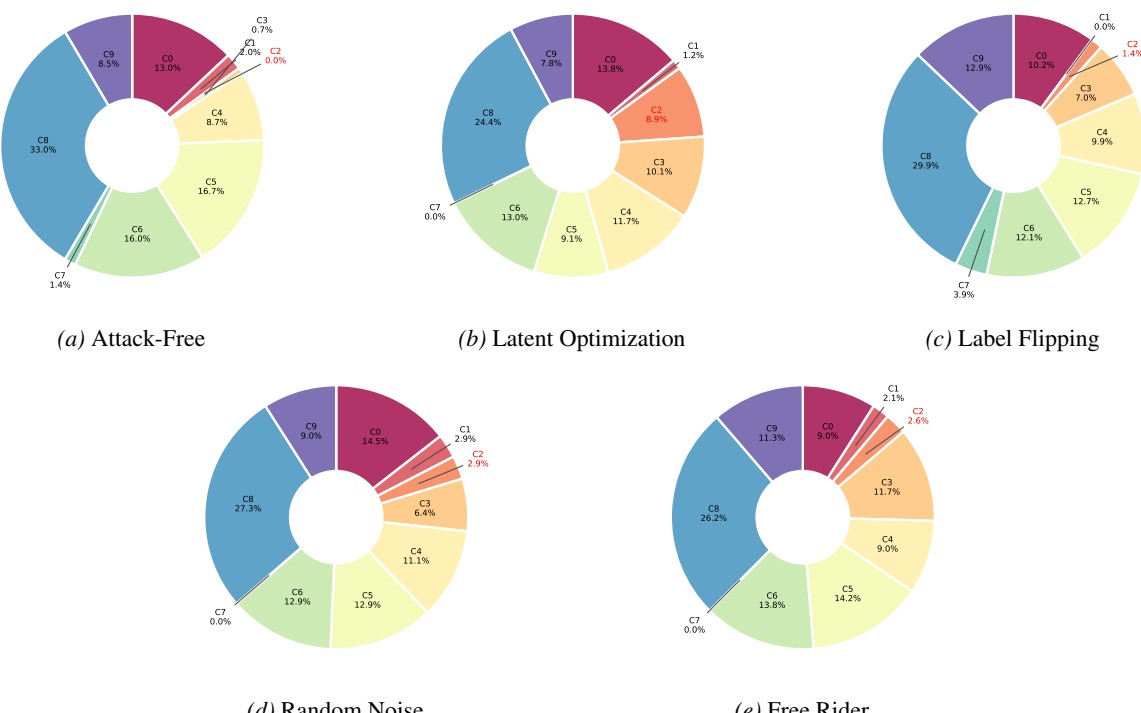

*(a)* Attack-Free     *(b)* Latent Optimization     *(c)* Label Flipping

*(d)* Random Noise     *(e)* Free Rider

*Figure 9.* **Client-level data attribution under FashionMNIST with WRN-28×10.** Per-client attribution shares under different attack settings. Colors indicate client identities and are consistent across panels.

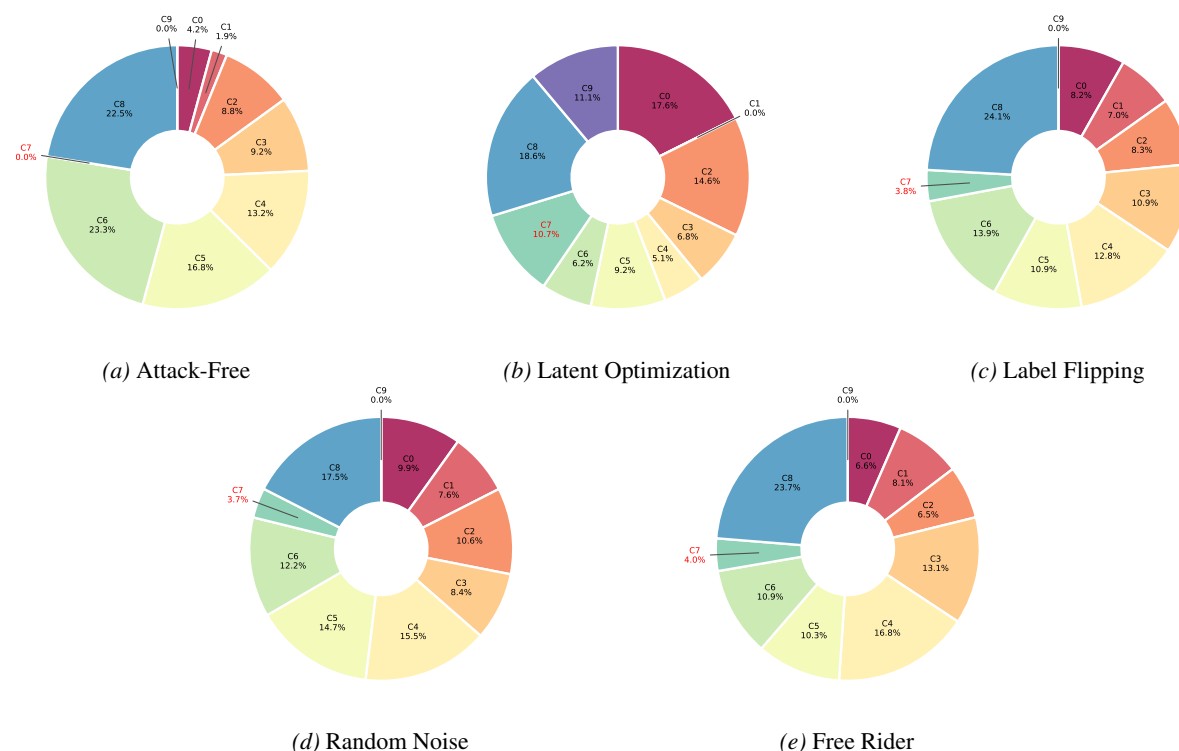

*(a)* Attack-Free     *(b)* Latent Optimization     *(c)* Label Flipping

*(d)* Random Noise     *(e)* Free Rider

*Figure 10.* **Client-level data attribution under FashionMNIST with VGG16_BN.** Per-client attribution shares under different attack settings. Colors indicate client identities and are consistent across panels.

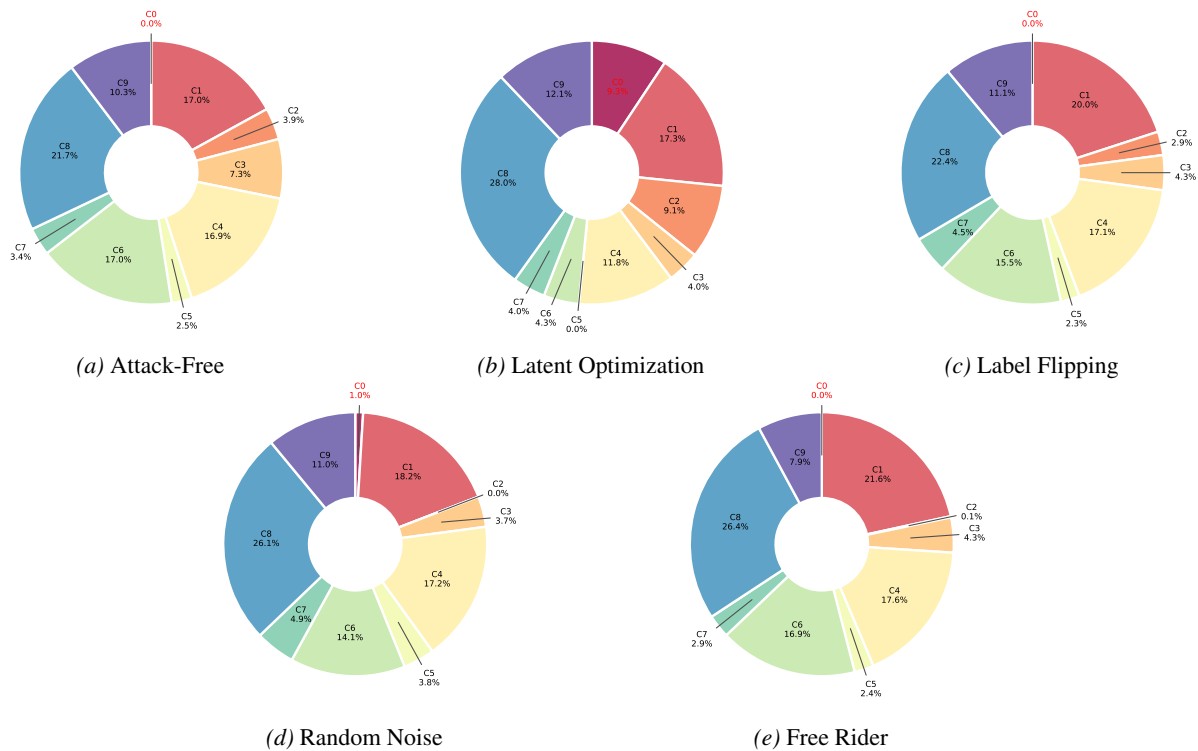

*Figure 11.* **Client-level data attribution under SVHN with ResNet-18.** Per-client attribution shares under different attack settings. Colors indicate client identities and are consistent across panels.

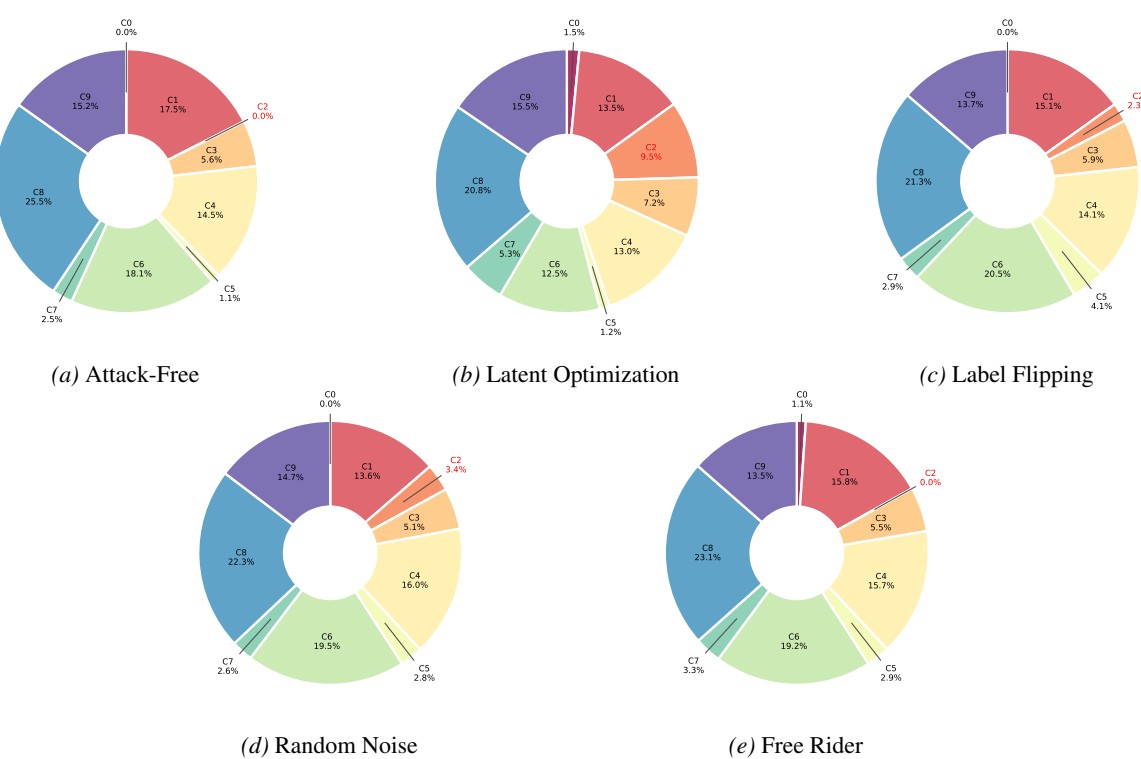

*Figure 12.* **Client-level data attribution under SVHN with WRN-28×10.** Per-client attribution shares under different attack settings. Colors indicate client identities and are consistent across panels.

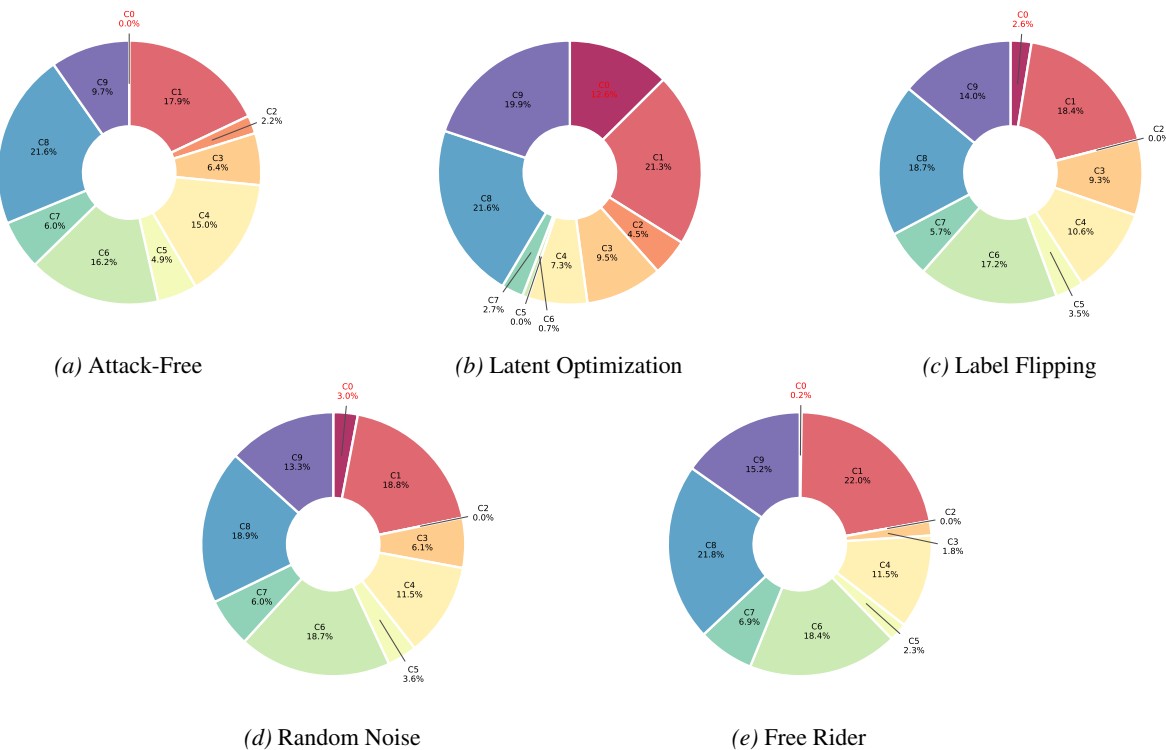

*Figure 13.* **Client-level data attribution under SVHN with VGG16-BN.** Per-client attribution shares under different attack settings. Colors indicate client identities and are consistent across panels.

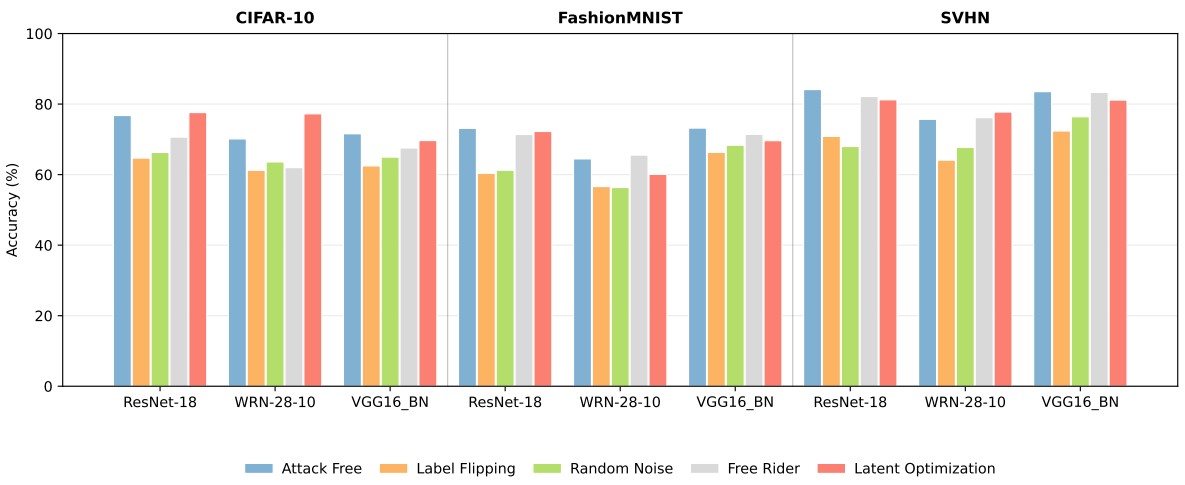

*Figure 14.* **Global model accuracy across datasets and architectures under different attack methods (FedSV).** We report test accuracy (%) for three model architectures on CIFAR-10, FashionMNIST, and SVHN. Each dataset–model pair is shown as a group of bars, with different colors indicating attack variants.

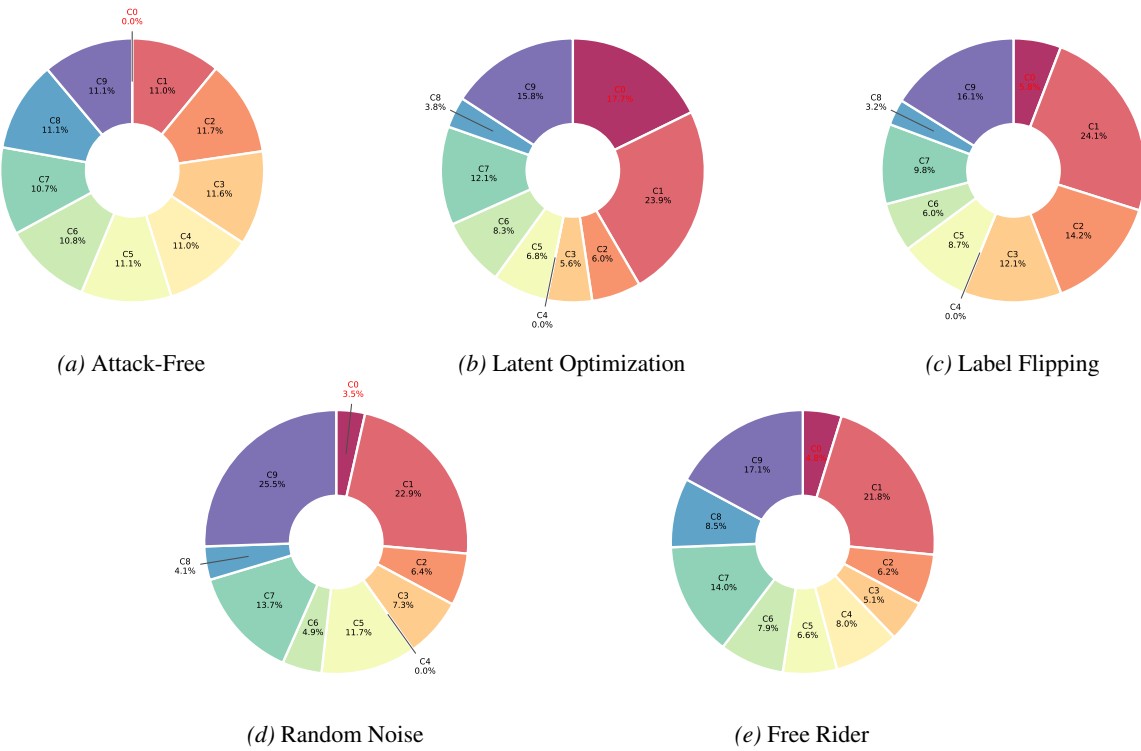

*(a)* Attack-Free       *(b)* Latent Optimization       *(c)* Label Flipping

*(d)* Random Noise       *(e)* Free Rider

*Figure 15.* **Leave-one-out (LOO) client-level data attribution under CIFAR-10 with ResNet-18.** Per-client attribution shifts under different attack settings. Colors indicate client identities and remain consistent across panels.

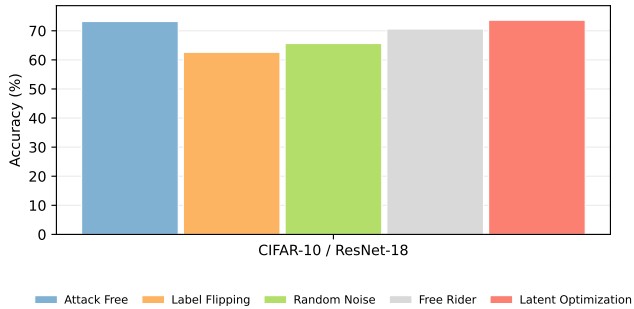

*Figure 16.* Global model accuracy under CIFAR-10 with ResNet-18 under different attack methods (LOO).

