# OpenReview forum: "On the Fragility of Data Attribution When Learning Is Distributed"
_ICML.cc/2026/Conference — ICML 2026 regular_

### Official Review · Reviewer_HhhA · 2026-03-09

**Soundness:** 3
**Presentation:** 3
**Significance:** 3
**Originality:** 4
**Overall Recommendation:** 4
**Confidence:** 3

**Summary:**

This paper proposes an attribution-first attack model so that FL clients can use it to inflate their attribution scores and gain unjustified advantages. They do so by ensuring their submitted gradient (from their synthetic data) at round $t$ is close to the reference global gradient (inferred from the parameters received at rounds $t-1$ and $t$) and has a small loss with regards to the reference global parameters received at round $t$. They empirically validate their methods on three vision benchmarks and compared with other attacks like Free-Rider.

**Compliance With Llm Reviewing Policy:**

Affirmed.

**Final Justification:**

The author's rebuttal has addressed all my major concerns and I have increased my rating from 3 to 4. I believe including these additional results in their revision would greatly improve the paper.

**Key Questions For Authors:**

- Refer to **Weaknesses**.
- What is the synthetic batch size $|B_s|$ used in the experiments and how should one set it in practice? How do attribution scores change when it changes?
- In Fig. 3, how are the clients’ data assigned?

**Limitations:**

yes

**Strengths And Weaknesses:**

Strengths:
- This is (maybe one of) the first paper that considers attribution-first attack, where the primary goal is to increase the attribution score. This is realistic and significant in today’s FL settings.
- The description is clear and easy to understand.
- The proposed method seems new and interesting.


Weaknesses:
- I suppose the quality of initial synthetic data $(\tilde{x}, \tilde{y}) = \mathrm{Dec}(z)$ is important (otherwise it suffices to just use a random $\tilde{x}$). What decoder do the authors use in their experiments? Are such decoders easy to find in practice?
- The paper considers a rather extreme non-iid case, where each client observes a subset of one class. In such cases, diversity (in terms of labels) would matter a lot to the attribution scores. Would the proposed method be less effective when the clients’ data are still non-iid but less extreme? For example, each client owns $50$% of one class and $50$% of all other classes evenly.
- Based on the above, there seems to be simpler alternatives for the attackers. For example, they can just use the benign reference gradient $g_t^{\textdagger}$ as $g_t^{(s)}$}, which considers all classes (from the past iteration), instead of using gradients from synthetic data that are optimized to align with the reference gradient. This would still work based on the theoretical analysis in Sec. 3.4. I think the paper would benefit from explaining why such alternatives are not chosen.

I would be happy to raise my score if the above concerns are addressed properly.

---

> ### Author Rebuttal · Authors · 2026-03-30
>
> Thank you for the thoughtful comments.
>
> >Weakness 1: quality of initial synthetic data
>
> **Response:** We use a pre-trained Vanilla VAE decoder to map latent codes into synthetic samples.
>
> The pre-trained VAE decoder can be easily obtained from public open-source repositories. Therefore, the use of a pre-trained decoder does not constitute a restrictive assumption.
>
> We agree that the quality of the decoder is important. The decoder is not assumed to be arbitrary or perfect, but rather serves as a practical generative prior.  To address the concern, we further replace the VAE with a BetaVAE decoder, which produces less flexible samples. The attack remains consistently effective under this change. This indicates that the attack is robust to the specific choice of decoder.
>
> | Target Rank | Decoder    | Attribution | Rank | Accuracy |
> |-------------|------------|-------------|------|----------|
> | 6           | VanillaVAE | 0.1087      | 4    | 73.94    |
> | 6           | BetaVAE    | 0.0924      | 5    | 74.61    |
> | 10          | VanillaVAE | 0.1327      | 4    | 73.39    |
> | 10          | BetaVAE    | 0.1263      | 5    | 72.02    |
>
> >Weakness 2: non-IID case
>
> **Response:** Following the reviewer's insightful suggestion, we conduct an additional experiment under a new non-IID distribution, where each client owns 55\% of one dominant class and 5\% from each of the remaining 9 classes (55+9*5=100). With the suggested non-IID, the malicious client still achieves higher attribution values and competitive attribution ranks. This demonstrates that the proposed method does not rely on extreme label sparsity to be effective.  Compared with the original setting, the attribution gains are moderately reduced in the new non-IID setting. This behavior is consistent with our design: when label diversity increases, the attacker naturally shifts from a missing-class completion to an imbalance correction mechanism.
>
> | Original rank    | Value  | Rank | Acc   |
> |------------------|--------|------|-------|
> | 2 (Original)     | 0.1522 | 1    | 73.59 |
> | 2 (new Non-IID)  | 0.1473 | 2    | 75.04 |
> | 6 (Original)     | 0.1063 | 4    | 76.94 |
> | 6 (new Non-IID)  | 0.0849 | 5    | 74.62 |
> | 10 (Original)    | 0.1547 | 4    | 75.39 |
> | 10 (new Non-IID) | 0.1168 | 6    | 75.88 |
>
> >Weakness 3: alternatives of attacks
>
>  We agree that directly using a benign reference gradient is a meaningful alternative to attacks. We have conducted an additional experiment to evaluate this baseline.
>
> The new baseline indeed improves the malicious client’s attribution. This confirms the reviewer’s intuition that alignment with the global training direction can increase attribution scores. However, this baseline is significantly outperformed by our method. While the alternative is valid, our approach goes beyond simple reference-direction alignment and instead exploits attribution through a more structured mechanism.
>
>
> | Original Rank | Method      | Attribution Value | Attribution Rank | Accuracy |
> |---------------|-------------|-------------------|------------------|----------|
> | 6             | Direct Ref  | 0.0871            | 5                | 73.88    |
> | 6             | Full Method | 0.1011            | 4                | 73.94    |
> | 10            | Direct Ref  | 0.1053            | 7                | 72.63    |
> | 10            | Full Method | 0.1627            | 4                | 75.39    |
>
>
> >Question 1: batch size
>
> In our experiments, for each target label, the attacker generates a batch of synthetic samples with a fixed size of 64.
>
> Increasing the batch size generally strengthens the attack, but also increases computational cost. To verify the effect, we vary the batch size across {16, 32, 64, 128}.  Our results show a consistent trend: increasing the synthetic batch size leads to stronger manipulation of data attribution scores. This indicates that larger synthetic batches result in more influence on the attribution computation. However, increasing from 64 to 128 results in a substantial cost with marginal benefit.
>
> | Original rank | Batch size | 16    | 32    | 64    | 128   |
> |---------------|------------|-------|-------|-------|-------|
> | 6             | Value      | 0.062 | 0.081 | 0.098 | 0.107 |
> | 6             | Rank       | 6     | 5     | 4     | 4     |
> | 6             | Acc   \%   | 76.70 | 76.95 | 77.20 | 77.28 |
> | 10            | Value      | 0.041 | 0.072 | 0.121 | 0.138 |
> | 10            | Rank       | 9     | 7     | 4     | 3     |
> | 10            | Acc   \%   | 76.48 | 76.86 | 77.05 | 77.12 |
>
> >Question 2: Figure 3
>
> In Figure 3, we adopt an imbalanced non-IID setting, rather than an extreme one-class-per-client setting. Specifically, by default, each client is assigned random data from multiple classes.
> For each run, the dataset is re-partitioned, ensuring that Figure 3 isolates the randomness. Following the reviewer's suggestion, we have conducted an experiment in a different non-IID setting (see response to Weakness 2).

---

> > ### Author Rebuttal · Reviewer_HhhA · 2026-04-01
> >
> > I thank the authors for their responses and additional experiments. **These have addressed all my major concerns.** Now I have a positive opinion about this paper and would increase my score to 4.
> >
> > I still have two small questions that require only explanations but not further experiments:
> > - Does the pretrained VAE decoder need to be from data of the same/similar domains?
> > - Could the author give some possible reasons why their methods have better performance than simply using the benign reference gradient (because in the rebuttal you only say “more structured”? I think a possible reason is that the gradients used by the paper come from real/almost real data so it can capture some additional data-specific information that the benign gradient does not have.
> >
> > It will be great if the authors could clarify the above two subtle points in their revision/final responses!

---

> > > ### Author Response · Authors · 2026-04-01
> > >
> > > Dear Reviewer HhhA,
> > >
> > > Thank you for reading through our response.
> > > We are glad to have addressed your major concerns. It is our pleasure to provide further explanations.
> > >
> > > >Question 1: Does the pretrained VAE decoder need to be from data of the same/similar domains?
> > >
> > >
> > > **Response:** The pretrained decoder does not need to come from the same or a similar domain as the data used in FL. In the experiment, the VAE decoder is pre-trained on face data (CelebA), which differs from the data used in FL: CIFAR-10 (object), FashionMNIST (clothing), and SVHN (digits). While the decoder serves as a generative prior, using one from the same domain can benefit the attack. In our experiments, we deliberately avoid using a decoder from the same domain because we want to avoid potential concerns about data leakage in the setting.
> > >
> > > >Question 2: Could the author give some possible reasons why their methods have better performance than simply using the benign reference gradient (because in the rebuttal, you only say “more structured”? I think a possible reason is that the gradients used by the paper come from real/almost real data, so it can capture some additional data-specific information that the benign gradient does not have.
> > >
> > > **Response:** We thank the reviewer for the insightful explanation. Due to the limitation of words in the rebuttal, we were not able to provide a detailed explanation. We totally agree with this intuition raised by the reviewer. Our approach dynamically optimizes latent attack through synthetic data such that the resulting gradients satisfy multiple objectives simultaneously, including maximizing attribution gain and effectiveness. This is different from the straightforward attack based on the benign reference gradient, which could be noisy, batch-dependent, and not necessarily aligned with the global training trajectory.
> > >
> > > We will carefully integrate all comments and new experiments in the revision.
> > > We truly value your constructive feedback and wish you all the best.

---

### Official Review · Reviewer_dyH9 · 2026-03-11

**Soundness:** 4
**Presentation:** 4
**Significance:** 3
**Originality:** 4
**Overall Recommendation:** 5
**Confidence:** 3

**Summary:**

The authors consider the question of whether participators in a federated learning setting can engineer their samples in a way that will give them an outsized valuation, despite having only a small effect on the actual test loss.

Data valuations are often used to align the economic interests of data providers, giving providers an incentive to make their data over-valued and motivating research into what malicious providers might be able to do in various settings.

The authors focus on the case of federated learning, and consider a potential attack, where in each iteration, the malicious client generates artificial samples that are optimized to have loss gradients that push the model further in the direction it moved over the last iteration, while looking somewhat natural to basic outlier detections (e.g., similar gradient norms), and potentially on under-represented classes.

Because these artificial samples have loss gradients that are aligned with the path the model was taking anyways, their effect on training would be similar to a slight change in learning rate but would not have an outsized effect on the trained model or its accuracy, but would cause reasonable data valuations to give them credit for the contributions of other providers.

The authors verify this experimentally, and show that this attack can indeed cause a single client to be highly over-valued.

**Compliance With Llm Reviewing Policy:**

Affirmed.

**Final Justification:**

The authors addressed all of my questions / all the weaknesses that I raised in their rebuttal, and I have adjusted my recommendations accordingly.

**Key Questions For Authors:**

See questions above.

**Limitations:**

yes

**Strengths And Weaknesses:**

The paper is very well written, the approach seems novel and the techniques and experiments look sound.

My biggest concern is the motivation of value-hacking *in the federated learning setting*. This is primarily because I would typically not expect systems that rely too much on client-side computation to be reliable for credit assignment to begin with.

On this note, there are 4 questions that I felt were not adequately addressed in the submission, and addressing them more thoroughly would strongly improve the quality of the paper.
1. Is this setting sufficiently well-motivated? (i.e., are there real-world / realistic scenarios where one would want to compensate clients for their data when the learning is federated -- clearly data valuation and federated learning are important and well-established fields on their own, but is there enough motivation for their cross-over)
2. Does the attacker gain a significant advantage from this setting? If we give the attacker significantly more power, then our baseline for what they can achieve should also increase. If attackers optimized for the standard setting, are much less effective than the FL-attack when both attacks are compared in the FL setting, that would provide much stronger motivation for researching attacks in the FL setting.
3. Can a similar attack for a non-FL setting be extracted? (e.g., if $z$ and its corresponding decoded sample was optimized based on an independent training trajectory or given only a trained model, and then the resulting samples were added to a fresh model's training, would that also cause the data valuation on the fresh model to assign a high value to these synthetic samples? if so, then this framework could be applied in the "offline" setting where we are only allowed to add samples to the train-set and do not have write access to any training internals)
4. Can this attack be mitigated by simple counter-measures? The authors consider the some existing counter-measures. One simple counter-measure they overlook is requiring the clients in the federated learning to pre-commit to their samples (e.g., via a Merkle tree) and then having the server validate a small fraction of their computation. The proposed attack seems strongly dependent on generating the malicious sample live (i.e., the malicious samples used in step $t$ of the training depend on the model at the start of this step).

I feel like these are the main weakness of the paper.

---

> ### Author Rebuttal · Authors · 2026-03-30
>
> Thank you for your careful and constructive review. We are encouraged by your feedback.
>
>
> >Weakness 1: Crossover of data valuation and federated learning
>
> **Response:** This crossover is well motiveted and studied by several works[1,2,3] in literature. These works treat attribution as a core component of federated systems for fairness, incentive design, and robustness. Besides research papers, there are also real world scenarios: Cross-institution FL has been actively explored in domains such as healthcare, where data cannot be centralized but collaboration is essential. For example, multi-hospital federated learning collaborations, like NVIDIA EXAM, demonstrate real-world deployments involving multiple data holders. In such settings, once multiple parties jointly contribute to a model, questions of who contributed how much and how to allocate credit or responsibility naturally arise.
> When attribution is integrated into FL, it introduces incentives to manipulate perceived contribution rather than true utility. We will clarify the motivation more explicitly.
>
> >Weakness 2: attacker's advantage from FL setting
>
> **Response:** Yes. The attacker gains a significant advantage from the FL setting, because the attacker can directly manipulate local model updates, which is not possible in traditional centralized training. To verify whether such attacker capabilities can outperform standard attack baselines, we introduce Outlier Attack [4] as a new baseline. Outlier Attack employs adversarial examples to generate manipulated datasets
>
> As shown in the table, Outlier Attack is readily detected by geometry-based trimming (higher detection rates than random guessing). This is becuase Outlier Attack operates at the data level and does not account for the client-update-level attribution in FL. In contrast, our method remains effectively undetected.
>
> Therefore, effective attribution attacks in FL must be designed at the level of client updates and account for the underlying attribution mechanisms.
>
> | Method              | Precision | Recall | F1-Score |
> |---------------------|-----------|--------|----------|
> | Random Guess        | 0.10      | 0.10   | 0.10     |
> | Outlier Attack      | 0.41      | 0.36   | 0.38     |
> | Latent Optimization | 0.00      | 0.00   | 0.00     |
>
>
> >Weakness 3: similar attack for a non-FL setting
>
> Our attack is fundamentally tied to FL setting and its training dynamics.
> A key aspect of our method is that the attacker has iterative access to the evolving global model. In each round, the attacker can adapt its strategy by re-optimizing latent vectors and regenerating synthetic samples. This creates a feedback loop that allows the attack to align with the training trajectory.
>
> In contrast, the offline setting removes this interactive nature and disables direct manipulations of model updates. Without access to the training process, the attacker cannot adapt to the evolving model or directly influence how contributions are computed. In an offline setting, adding synthetic samples may affect model performance, but does not provide the same control over attribution outcome.
>
> >Weakness 4: simple counter-measures
>
> Following the reviewer’s suggestion, we introduce a defense based on pre-commitment and partial computation validation. Specifically, we assume that each client pre-commits to its local dataset, and the server randomly validates a small fraction (5\%) of the committed computation in each communication round. A client is rejected if its update fails the validation check.
>
> In the table below, we observe that partial validation provides limited detection capability. The malicious client continues to achieve high attribution values and maintains a competitive attribution rank across all settings. This is because the attack operates at the level of client updates and produces hybrid updates that are dominated by real data, aligned with the global optimization direction, and consistent in magnitude.
> The results demonstrate that simple countermeasures based on partial computation validation are insufficient to mitigate the proposed attack. This is because the attacker does not rely on injecting fully adversarial or anomalous samples, but instead generates updates that are consistent with benign training dynamics. Such attack can't be detected through small-fraction validation.
>
> | Target Rank | F1   | Attr. Value | Attr. Rank | Acc. (\%) |
> |-------------|------|-------------|------------|-----------|
> | 2           | 0.14 | 0.1396      | 2          | 70.91     |
> | 6           | 0.12 | 0.0918      | 5          | 73.84     |
> | 10          | 0.09 | 0.1342      | 5          | 71.62     |
>
> [1] Fair and efficient contribution valuation for vertical federated learning, ICLR 2024
>
> [2] Redefining Contributions: Shapley-Driven Federated Learning, IJCAI 2024,
>
> [3] PACE: Single-round Participant Amalgamation for Contribution Evaluation in Federated Learning, NeurIPS 2023
>
> [4] Adversarial Attacks on Data Attribution

---

> > ### Author Rebuttal · Reviewer_dyH9 · 2026-04-01
> >
> > The rebuttal resolved all of my concerns, by giving better motivation for the problem, explaining why the federated setting does indeed allow for qualitatively stronger attackers, and showing that simple mitigations fail to address this issue. If these are all added to the final version of the paper, that would make it much stronger in my opinion.

---

> > > ### Author Response · Authors · 2026-04-01
> > >
> > > Dear Reviewer dyH9,
> > >
> > > Thank you for your thoughtful and encouraging feedback. We are glad that our rebuttal has addressed your concerns and strengthened the motivation, setting, and empirical support.
> > >
> > > Your recognition means a great deal to us. We will carefully incorporate all clarifications and results into the final version to further improve the paper.
> > >
> > > We sincerely appreciate the time and effort you have devoted to providing such constructive and supportive comments, and we wish you the best.

---

### Official Review · Reviewer_5DKT · 2026-03-13

**Soundness:** 1
**Presentation:** 2
**Significance:** 3
**Originality:** 2
**Overall Recommendation:** 4
**Confidence:** 3

**Summary:**

This paper focused on the adversarial attack to data attribution score in a non-IID scenario such as federated learning. The paper points out an attack model where a single participant can manipulate the contributed dataset (through adding synthetic data) to raise the attribution value of its own and reahspe the relative attribution structure of others. Paper did some experiments to show that the method could also avoid degrading accuracy too much.

**Compliance With Llm Reviewing Policy:**

Affirmed.

**Final Justification:**

I am willing to increase the score to 4 since the author has addressed my major concern.

**Key Questions For Authors:**

What is the intuition that generating samples that are aligned with global progress could help produce a higher attribution score?

**Limitations:**

yes

**Strengths And Weaknesses:**

**Strength**:

Data attribution is getting more and more popular in downstream scenarios such as pricing, auditing, and governing. The robustness of the data attribution score is a pressing challenge and open question to the community. This paper could motivate the community to understand and find methods to mitigate the challenge.

Several papers touched on the topic, such as [1] and [2]. The paper proposes a better feature of the attack: less degraded accuracy, which tends to be the first try in this direction.

**Weakness**

The intuition that generating samples that are aligned with global progress could help produce a higher attribution score is not well justified. The newly generated samples seem to perform as a “momentum” to the current optimizer step rather than a valuable data sample. Another problem is that I am not sure if assuming real data D_r^{i*} can be obtained by the adversarial client is reasonable.

The presentation of the paper can be improved. One particular way is to have a diagram showing the attack method and the threat model.

A direct baseline could be [2] which is also a way that change dataset to affect the attribution score. The comparison between [2] and the method proposed in this paper could be useful.

[1] Yadav, C., Wu, R., & Chaudhuri, K. Influence Attributions can be Systematically Altered by Model Manipulation. In The 29th International Conference on Artificial Intelligence and Statistics.

[2] Wang, X., Hu, P., Deng, J., & Ma, J. W. (2024). Adversarial attacks on data attribution. arXiv preprint arXiv:2409.05657.

---

> ### Author Rebuttal · Authors · 2026-03-30
>
> We appreciate the reviewer's constructive feedback.
>
> >Weakness 1: The value of generated samples and the assumption of obtaining $D_r^{i*}$
>
> **Response:** To address the review's concern, we conduct an ablation study comparing (i) the full latent optimization method and (ii) an unconstrained variant that removes the structural objectives. If the observed effect were due to a momentum-like mechanism, the unconstrained variant would achieve comparable attribution gains. However, the unconstrained variant only yields limited improvements, while the full method consistently achieves significantly higher attribution values. The gap indicates that unstructured gradient reinforcement alone is insufficient to explain the effectiveness of our method.
>
> | Target Rank | Method | Attribution Value| Attribution Rank | Accuracy |
> |----------------------|-----------------|----------------------------|---------------------------|------------------------------|
> |            2          | Unconstrained   | 0.1365                     | 2                         | 73.02                        |
> |           2           | Full Method     | 0.1618           | 1               | 71.41               |
> |              10        | Unconstrained   | 0.0926                     | 8                         | 74.05                        |
> |             10         | Full Method     | 0.1682            | 4              | 76.59              |
>
>
> We remark that $D_r^{(i)}$ denotes the local dataset of client $i$.  For attacker $i^\*$, we use $D_r^{(i^*)}$ to represent the attacker's own existing data.
> Therefore, our threat model does not assume access to other clients' private data.
> This setting is the standard in cross-silo FL.
>
> >Weakness 2: presentation
>
> We have added a new diagram to demonstrate the attack method: https://anonymous.4open.science/r/Weakness_2-0D4B/Weakness_2.pdf.
>
> > Weakness 3: comparison with [2]
>
> Following the reviewer’s suggestion, we compare [2] with our method. We remark that [2] is originally designed in the setting of centralized training. However, the effectiveness does not transfer to FL setting.
>
> The objective of Shadow Attack [2] is to identify high-attribution data. However, in FL, the attacker does not directly submit data, but instead submits model updates. The attribution signal of individual data points is diluted by local training and further attenuated during aggregation.
>
> The limitation of the Outlier Attack [2] is more fundamental. Its objective is to maximize loss through adversarial perturbations. However, FL methods such as FedSV reward positive utility. As a result, outlier data often produce noisy and misaligned gradients that deviate from the global optimization.
>
> Overall, these results demonstrate that attribution manipulation in FL is fundamentally different from centralized settings. Existing data-level attacks are not directly applicable.
>
> | Target Rank | Method            | Attribution Value | Attribution Rank | Accuracy |
> |-----------------------------|----------------------------|----------------------------|---------------------------|------------------------|
> |                2             | Shadow Attack              | 0.1248                     | 3                         | 71.92                  |
> |                2             | Outlier Attack             | 0.0916                     | 5                         | 67.47                  |
> |              2               | Latent Optimization Attack | 0.1522                     | 1                         | 73.59                  |
> |            10                 | Shadow Attack              | 0.1129                     | 6                         | 72.75                  |
> |            10                 | Outlier Attack             | 0.0798                     | 8                         | 68.85                  |
> |           10                  | Latent Optimization Attack | 0.1547                     | 4                         | 74.39                  |
>
> > Question 1: the intuition
>
> The intuition is that data attribution, such as FedSV, assigns higher scores to updates that advance the global model along its current optimization, rather than considering whether the update is malicious.
> Specifically, attribution is computed based on utility gains. Under this mechanism, updates that are aligned with the global progress direction are naturally interpreted as making positive contributions. Intuitively, such alignment reinforces the direction in which the model is improving, and therefore consistently produces marginal utility gains. In contrast, if a client’s update is not aligned with the global progress, it may contribute less effectively to the current optimization trajectory, or even introduce conflicts. As a result, its marginal utility becomes smaller, leading to a lower attribution score under utility-based evaluation.
>
> We have conducted additional ablation to verify the intuition (weakness 1).

---

> > ### Author Rebuttal · Reviewer_5DKT · 2026-04-03
> >
> > Thank you for the suggestion; the reply addressed my key questions. I will raise my score to 4.

---

> > > ### Author Response · Authors · 2026-04-03
> > >
> > > Dear Reviewer 5DKT,
> > >
> > > We sincerely appreciate your acknowledgement and are grateful that our response has addressed your concerns. Your positive feedback is truly encouraging to us.
> > >
> > > We will ensure that these improvements are clearly incorporated into the final version of the paper.
> > >
> > > Thank you again for your thoughtful consideration and support, and we wish you all the best.

---

### Decision · Program_Chairs · 2026-04-30

**Decision:**

Accept (regular)

**Comment:**

This paper investigates adversarial attacks on data attribution in distributed/federated learning settings. The authors proposed an attack where malicious clients produce adversarial samples that receive inflated attribution scores. The proposed attack is empirically validated.

Overall, reviewers agreed that this is a well-written paper with a novel method and sound experiments. There were concerns around the motivation of the proposed problem setting and detailed comparison with existing adversarial attack methods in this area. However, these concerns were well-addressed in the rebuttal.